# The Molecular Mechanism of Relatively Low-Temperature-Induced Broccoli Flower Bud Differentiation Revealed by Transcriptomic Profiling

Wenchen Chai [1,2,†], Xia He [1,†], Boyue Wen [1], Yajie Jiang [1], Zixuan Zhang [1], Rui Bai [1], Xinling Zhang [3], Jin Xu [1,2], Leiping Hou [1], Meilan Li [1,2] and Jing Zhang [1,*]

[1] College of Horticulture, Shanxi Agricultural University, Taigu 030801, China; 15935485975@163.com (M.L.)
[2] Key Laboratory of Innovation and Utilization for Vegetable and Flower Germplasm Resources in Shanxi, Taiyuan 030000, China
[3] Institute of Vegetables, Zhengzhou 450000, China; zxl99988@126.com
[*] Correspondence: zhangjing_1010@126.com
[†] These authors contributed equally to this work.

**Abstract:** Broccoli (*Brassica oleracea* L. var. *italica*) has a large, edible green flower head, which is one of its critical economic traits. A prerequisite of broccoli flower head formation is flower bud differentiation (FBD). Low-temperature treatment is an effective way to induce FBD in broccoli. However, the molecular mechanism underlying low-temperature-induced broccoli FBD remains largely unclear. In this study, using broccoli cultivar Zhongqing 10 as the experimental material, we investigated the effects of low-temperature treatment on FBD by comparing the plants grown at low temperatures (17 °C/9 °C, 16 h/8 h) with the control plants grown under normal temperature conditions (25 °C/17 °C, 16 h/8 h). After 15 days of different temperature treatments, the flower buds of the plants growing under the low-temperature condition started to differentiate. However, the control plants remained in the vegetative growth stage, indicating that low temperature successfully induced flower bud formation. Subsequently, a global transcriptomic analysis was conducted to detect the differentially expressed genes (DEGs) during low-temperature-induced FBD in broccoli. A total of 14 DEGs in five phytohormone signaling pathways, 42 DEGs in nine transcription factor families, and 16 DEGs associated with the floral development pathways were identified. More DEGs were present in the auxin signaling pathway than in other phytohormone signaling pathways, which indicated that the auxin signaling pathway played a critical role in modulating low-temperature-induced FBD in broccoli. Furthermore, four TF classes, including bZIP, GCM domain factors, MADS-box factors, and C2H2 zinc finger factors, possessed enriched motifs, indicating that their closely related DETFs *ABI5*, *HY5L*, *WRKY11*, *WRKY15*, *WRKY22*, *SOC1*, *AGL8*, *FLC*, *SPL8*, and *SPL15* may be directly involved in the transcription regulation of broccoli FBD. This study provides an important basis for further investigation of the molecular regulatory mechanism of broccoli flower development under low temperatures.

**Keywords:** broccoli; flower bud differentiation (FBD); relatively low temperature; transcriptomic profiling





## 1. Introduction

Broccoli (*Brassica oleracea* L. var. *italica*) is a member of the cruciferous vegetable family. It is consumed around the world and is a mild-climate species originating in the Mediterranean region [1,2]. It is a highly nutritious vegetable crop, valued primarily for its flower head, which contains high quantities of vitamins, minerals, and anti-cancer substances such as sulforaphane and glucosinolates [3,4]. Broccoli leaves also contain nutrients such as protein, fiber, and phenolic compounds [5]. Therefore, broccoli has substantial nutritional and dietary value and has a higher economic value than similar vegetables. Broccoli is a vernalizing plant, meaning that its yield can only be achieved when the plant body undergoes vernalization, during which the advancement of FBD will determine subsequent flower bud development and quality.

Significantly, inflorescence development of cruciferous vegetables not only affects reproduction, but also affects the yield and quality of the commodity organs. The timing and quality of FBD also affect inflorescence development [6,7].

Broccoli is a cool-season plant that prefers full sunlight; however, the light requirement for its vernalization is not strict. The plants vernalize properly at low temperatures, and their flower buds differentiate after a certain period of vegetative growth. The vegetative stage depends on the particular broccoli cultivar and on the time of cultivation. In addition, the major environmental factor affecting the development of inflorescences is temperature [8–10], and low temperature is an essential factor determining the transition from vegetative growth to reproductive growth [11]. This transition is a crucial physiological change in broccoli plants.

FBD is a physiological and morphological indicator of the transition from vegetative to reproductive growth and is a response to diverse endogenous and exogenous signals that lead to flowering [12,13]. Present studies indicate that there are six regulatory pathways of FBD in *Arabidopsis thaliana*, namely, the vernalization pathway, photoperiod pathway, gibberellin (GA) pathway, autonomic pathway, temperature pathway, and age pathway [14–16]. These pathways are independent but cross-linked to form a precise network for regulating floral development. For the cruciferous plants, low temperatures and long light periods are essential for FBD [17]. Moreover, the transcription regulation of gene expression in the vernalization pathway plays a significant role in the cruciferous plant FBD. *FLOWERING LOCUS C* (*FLC*), a central regulator of the vernalization pathway, is mainly expressed in the shoot apical meristem (SAM) and vascular tissues. In addition, *FLC* directly inhibits the expression of *FLOWERING LOCUS T* (*FT*) in leaves, as do *SUPPRESSOR OF OVEREXPRESSION OF CONSTANS 1* (*SOC1*) and *FLOWERING LOCUS D* (*FD*) in the SAM [18–20]. *CONSTANS* (*CO*) and *FT* are the most researched genes in the photoperiod pathway. Zinc finger transcription factors (TFs) encoded by *CO* play a key role in regulating FBD in response to photoperiod signals [17,21]. The prolongation of illumination time brings about the accumulation of *CO*, which directly activates the expression of *SOC1* and *FT* in the flowering pathway [22]. However, the functions of *CO-like* (*COL*) genes differ from those of *CO* [23]. Notably, in *Arabidopsis thaliana*, *COL4* and *COL9* are transcriptional inhibitors of flowering [24,25].

At present, the molecular mechanisms of FBD in *Arabidopsis thaliana*, *Brassica oleracea* L., *Brassica oleracea* L. var. *botrytis*, *Brassica juncea* cv. Varuna, and other cruciferous plants have been researched [15,26–29]. However, studies on broccoli FBD mainly focus on physiology and cytology [30–32], and the molecular mechanism of broccoli FBD remains unclear.

In this study, the broccoli cultivar Zhongqing 10 was used as the research subject to investigate the effects of relatively low temperatures on broccoli FBD, and this differentiation process was observed and identified using a stereomicroscope and paraffin sections. Based on them, RNA-seq technology was used to analyze the differentially expressed genes (DEGs) after different treatment durations at relatively low temperatures, and the key genes regulating broccoli FBD were sifted. In addition, the regulation mechanism of broccoli FBD at relatively low temperatures was explored, which provided an important basis for high-quality and high-yield broccoli cultivation and enhanced our understanding of the molecular mechanism of broccoli floral development.

## 2. Materials and Methods

### 2.1. Plant Materials and Treatment

The broccoli cultivar used was Zhongqing 10, and the seeds were purchased from China Vegetable Seed Technology (Beijing) Co., Ltd. (Beijing, China). The seeds were soaked in water for five hours and then sown into a 50-hole substrate tray for germination and seedling growth in a sunlit greenhouse (maximum daytime temperature: 30 °C and minimum nighttime temperature: 15 °C) for routine management.

Five-leaf broccoli seedlings of uniform size were placed in incubators for growth. In the control group, the seedlings were incubated under a set photoperiod [16 h of light at (25.0 ± 1) °C and 8 h of dark at (17.0 ± 1) °C] and designated as CK. In the relatively low-

temperature group (the treatment group), the seedlings were also incubated under a set photoperiod [16 h of light at (17.0 ± 1) °C and 8 h of dark at (9.0 ± 1) °C] and designated as V.

### 2.2. Paraffin Sectioning

The morphological changes of the broccoli shoot tips were observed by Olympus SZX16 stereomicroscopy (Olympus, Shinjuku-ku, Tokyo, Japan), to define the developmental stages. In the first ten days, six plants were observed every two days. After ten days of treatment, six plants were observed every day. This process was repeated three times. The shoot tips of the two groups treated for 0 days, 8 days, 14 days, 15 days, and 22 days were photographed. When the tissues at a specific developmental stage were discovered, they were fixed, sliced, and dewaxed according to the modified protocol described by Garcês and Sinha [33]. The tissue staining methods were as follows: 0.1% toluidine blue staining for 11 min, distilled water rinsing for 2 min, deionized water washing for 2 min, deionized water washing for another 2 min, 70% (*v/v*) ethanol dehydrating for 1 min, 85% (*v/v*) ethanol dehydrating for 1 min, 100% ethanol dehydrating for 2 min, xylene soaking for 3 min, and xylene soaking for another 3 min. Finally, the slides were placed in neutral balsam and observed and photographed using the Olympus IX81 microscope (Olympus, Shinjuku-ku, Tokyo, Japan).

### 2.3. Sampling for Transcriptomic Analysis

The broccoli shoot tips treated at relatively low temperatures for 0 days, 8 days, 14 days, 15 days, 22 days, and 15 days under the control condition were sampled and designated V0, V8, V14, V15, V22, and CK, respectively. At each time point, 24 shoot tips with a length of about four millimeters to five millimeters and containing one or two leaf primordium were collected from 24 individual plants and mixed to form three biological replicates. Samples (0.1 g) were taken for RNA extraction. Then, these samples were instantly frozen in liquid nitrogen and deposited at −80 °C for later use.

### 2.4. RNA-Seq and Differential Expression Analysis

Total RNA extraction and RNA-seq analysis were accomplished by Biomarker Technologies Co., Ltd. (Beijing, China), and clean reads were mapped to the cabbage genome sequence [https://www.ncbi.nlm.nih.gov/assembly/GCA_900416815.2/ (accessed on 24 December 2019) using HISAT2 [34]. FPKM was used for gene/transcript level quantification. Based on the raw count data, differential expression analysis between samples was carried out by the DESeq2 R package (1.16.1) [35]. Genes with |log2FC| ≥ 1, a false discovery rate of (FDR) < 0.01, and adjusted *p*-value ≤ 0.01 were defined as DEGs.

### 2.5. Motif Overrepresentation ESTIMATION

Motif overrepresentation of the promoters of DEGs from CK compared with V14 and V15 was estimated by the ESDEG tools [https://github.com/ubercomrade/esdeg (accessed on 16 November 2023)] [36]. Since the *Arabidopsis thaliana* motifs are quite good for all cruciferous plants, motifs of known TFs were compiled from the JASPAR database with *Arabidopsis thaliana*.

### 2.6. Quantitative Real-Time Polymerase Chain Reaction (qRT-PCR)

To test and verify the RNA-seq results, qRT-PCR was carried out using gene-specific primers (Supplementary Table S1) for ten genes related to broccoli FBD. Specific primers were designed using the Primer 3 software according to the gene sequence. The expression of the tested gene was determined using 18S RNA as an internal reference. The total RNA of each sample was extracted from V0, V8, V14, V15, V22, and CK according to the instructions of the plant RNA extraction kit [Tiangen Biotech (Beijing) Co., Ltd., Beijing, China]. First-strand complementary deoxyribonucleic acid (cDNA) was obtained using the PrimeScript™ RT Reagent Kit (Takara, Dalian, China). Subsequently, qRT-PCR was performed in an Applied Biosystems® 7500 Fast Real-Time PCR System (ABI, Foster, CA, USA) using the

TB Green® Premix Ex Taq™ kit (Takara, Dalian, China). qRT-PCR was conducted in a 20 μL reaction mixture containing 10 μL of 2 × TB Green Premix EX Taq, 2.0 μL of diluted cDNA, 0.4 μL of 50 × ROX Reference Dye II, 0.4 μL of forward primers (10 μM), 0.4 μL of reverse primers (10 μM), and sterile double-distilled water (ddH$_2$O). The thermal cycling parameters were set as follows: predenaturation at 94 °C for 30 s, 95 °C for 30 s, 55 °C for 30 s, and 72 °C for 30 s for 40 cycles. Each experiment was conducted in triplicate with three biological replicates, and the relative expression level of genes was calculated using the $2^{-\Delta\Delta CT}$ method [37].

## 3. Results

### 3.1. Morphological Changes at Different Stages of FBD

Broccoli usually grows appropriately in areas with an average temperature below 18 °C, and the FBD and formation of the flower heads in broccoli also require low temperatures. When the temperature is too high, the broccoli heading ability is significantly reduced, and the plants remain in vegetative growth [4]. In this research, the changes in the broccoli shoot tips were observed under a stereomicroscope. The observations revealed that FBD began at 15 d in the treatment group (17 °C/9 °C). In contrast, in the control group (25 °C/17 °C), the broccoli shoot tips remained in the vegetative growth stage, with a small volume and a small surface area surrounded by the triangularly protruding leaf primordium that was spirally differentiated around the central axis. Moreover, the growth cones were also in the vegetative growth stage at 0–13 d after treatment in the treatment group. However, the flower buds entered the transitional stage on the 14th day of treatment at relatively low temperatures. In this stage, the growth cones protruded, increased in volume, and could hardly be covered by the leaf primordium. The flower buds began to differentiate on the 15th day of treatment at relatively low temperatures. The growth cone protrusions became flat and widened, and their volume and surface area increased. More importantly, protrusions appeared on the growth cones. On the 22nd day of treatment at relatively low temperatures, flower primordium differentiation began in the lateral direction, and protrusions around the growth cones were prominent (Figure 1).

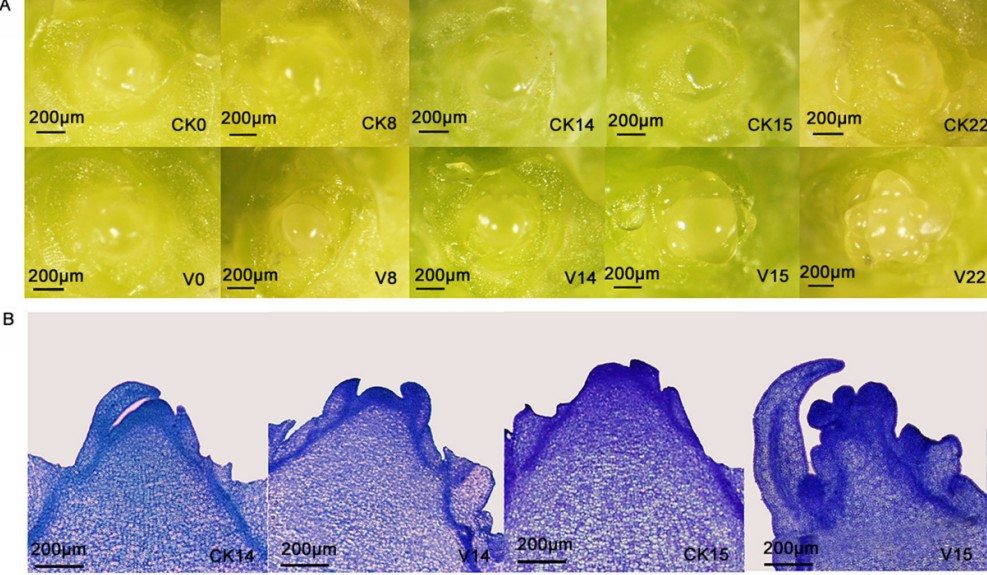

**Figure 1.** Broccoli shoot tip phenotype after different treatment durations under relatively low temperatures. (**A**) Stereomicrograph. (**B**) Paraffin sectioning micrographs at the critical stage of broccoli FBD. CK0, CK8, CK14, CK15, and CK22 are the broccoli shoot tips treated under a photoperiod of 16 h/8 h and temperature of 25 °C/17 °C for 0 days, 8 days, 14 days, 15 days, and 22 days. V0, V8, V14, V15, and V22 are the broccoli shoot tips treated under a photoperiod of 16 h/8 h and temperature of 17 °C/9 °C for 0 days, 8 days, 14 days, 15 days, and 22 days.

### 3.2. Evaluation of RNA-Seq Results

The effect of 17 °C/9 °C on the broccoli shoot tip transcriptomic changes was revealed by RNA-seq. More than 93.00% of the clean reads, including unique and multimapped reads, from all 18 libraries were mapped to the reference genome. The high proportion of mapped genes indicated that the sequencing libraries and reference genome were suitable for further analysis. The unique mapped reads accounted for over 90.00% of the total mapped reads from all 18 libraries (Table S2). In addition, the correlation coefficients among the biological replicates were more significant than 0.97 in all the samples (Figure S1), suggesting close correlations among the three biological replicates for each treatment, which could be used for further data analysis.

### 3.3. Screening and Analysis of DEGs

DEGs were compared in six comparisons, namely, V0 compared with V8, V0 with V14, V0 with V15, V0 with V22, CK with V14, and CK with V15, and the results are shown in Figure 2A and Table S3. In total, downregulated genes were more than upregulated genes during FBD at relatively low temperatures. Additionally, 18 upregulated DEGs were shared by V0 compared with V8, V14, V15, and V22, while only 142, 229, 141, and 85 DEGs were upregulated in the four comparisons, respectively (Figure 2B). In addition, 312 downregulated DEGs were shared by V0 compared with V8, V14, V15, and V22, while only 675, 66, 205, and 122 DEGs were downregulated in the four comparisons, respectively (Figure 2C). A total of 183 upregulated DEGs were shared by CK compared with V14 and V15, while 722 and 97 DEGs were only upregulated in one of the two comparisons, respectively (Figure 2D). Moreover, 1434 downregulated DEGs were shared by CK compared with V14 and V15, while 785 and 264 genes were only downregulated in one of the two comparisons, respectively (Figure 2E).

To gain insight into the dynamics of gene expression changes during FBD at relatively low temperatures, *k*-means clustering was used to analyze the fragments per kilobases per million reads (FPKM) data from V0, CK, V8, V14, V15, and V22. Two different kinetic clusters of coexpressed genes were obtained from V0, V8, V14, V15, and V22, all of which had significant transcriptional bursts during FBD under different treatment durations at relatively low temperatures (Figure 3A). The expression levels of 551 genes in Cluster 1 gradually downregulated as FBD progressed, indicating that they were negatively associated with broccoli FBD (Figure 3A, Table S4). The TF *FLC* also showed a negative correlation with broccoli FBD. Conversely, 316 genes in Cluster 2 displayed increased expression levels as FBD progressed, indicating that they were positively associated with FBD (Figure 3A, Table S4). The genes mentioned above encode multiple TFs, such as *AGAMOUS-LIKE8* (*AGL8*) and *SQUAMOSA PROMOTER BINDING PROTEIN-LIKE8* (*SPL8*), which showed correlations with broccoli FBD and might play crucial roles in regulating the expression levels of genes related to broccoli FBD.

Meanwhile, two different kinetic clusters of coexpressed genes were also obtained from CK, V14, and V15, all of which had significant transcriptional bursts during FBD under different treatments for different days (Figure 3B). In Cluster 1, the expression levels of 1591 genes gradually downregulated as FBD progressed, so they were negatively associated with FBD. Additionally, the expression levels of *COLs*, *EARLY FLOWERING 4* (*ELF4*), *FRIGIDA-LIKE* (*FRL*), and *GA2-oxidase 2* (*GA2ox2*) were negatively correlated with broccoli FBD. Instead, the expression levels of 468 genes in Cluster 2 upregulated as FBD progressed, indicating that they were positively associated with FBD (Figure 3B, Table S5). These genes encoded multiple TFs, including *FT* and *SOC1*, that were positively correlated with broccoli FBD.

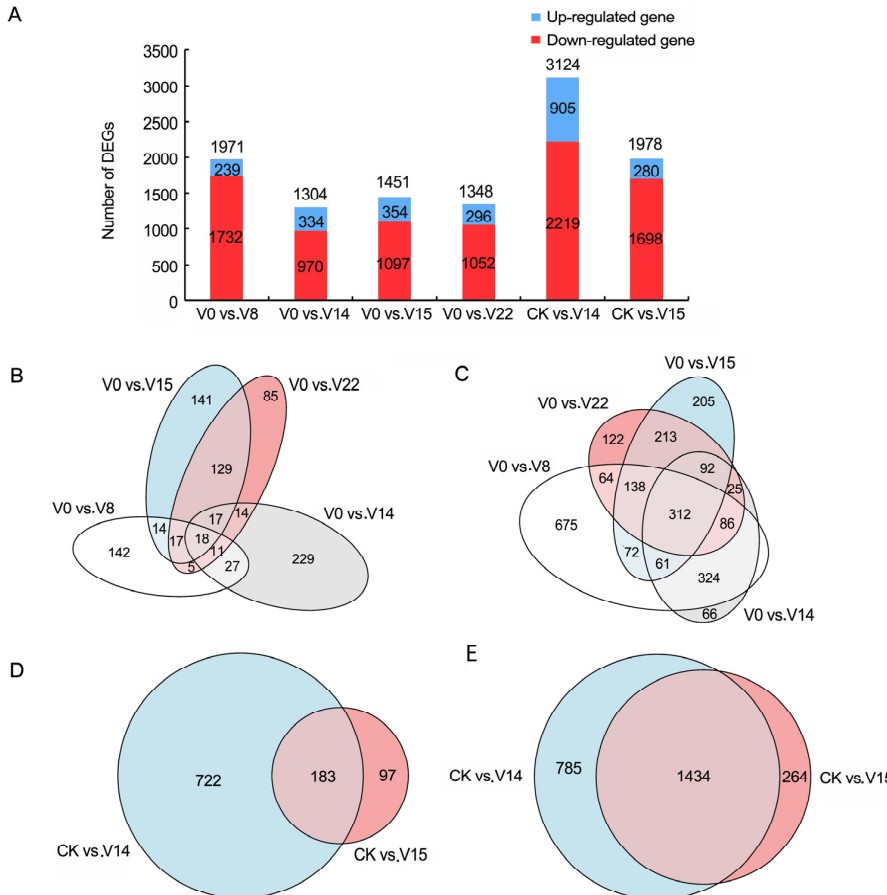

**Figure 2.** Comparative analysis of DEGs in broccoli shoot tips at relatively low temperatures. (**A**) The number of DEGs in V0 compared with V8, V14, V15, and V22, as well as CK with V14 and V15. (**B**) Venn diagram of the number of upregulated DEGs in V0 compared with V8, V14, V15, and V22. (**C**) Venn diagram of the number of downregulated DEGs in V0 compared with V8, V14, V15, and V22. (**D**) Venn diagram of the number of upregulated DEGs in CK compared with V14 and V15. (**E**) Venn diagram of the number of downregulated DEGs in CK compared with V14 and V15. V0, V8, V14, V15, and V22 represent the samples from the plants under 17 °C treatment for 0 days, 8 days, 14 days, 15 days, and 22 days, respectively; CK represents the samples from the plants under 25 °C treatment for 15 days.

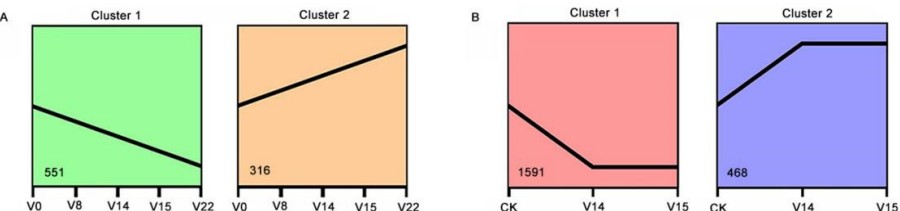

**Figure 3.** Coexpressed genes and their kinetic patterns in broccoli shoot tips at relatively low temperatures. (**A**) Two clusters of coexpressed genes and their dynamic patterns in broccoli shoot tips under different treatment durations at relatively low temperatures. (**B**) Two clusters of coexpressed genes and their dynamic patterns at the critical stage of broccoli FBD.

### 3.4. GO Enrichment Analysis and KEGG Pathway Analysis of Coexpressed DEGs

To reveal the biological processes (BPs), cellular components (CCs), and molecular functions (MFs) underlying the broccoli transcriptome, a gene ontology (GO) enrichment analysis was conducted on the kinetic clusters of coexpressed DEGs obtained from the two analysis groups. In Cluster 1 of Analysis Group 1 (V0 compared with V8, V14, V15, and V22), enriched

BP GO terms included the oxidation–reduction process, response to cold, and response to karrikin; enriched CC GO terms included the chloroplast thylakoid membrane, chloroplast envelope, and chloroplast stroma; and enriched MF GO terms had symporter activity, beta-glucosidase activity, and scopolin beta-glucosidase activity (Figure 4A). In Cluster 2 of Analysis Group 1 (V0 compared with V8, V14, V15, and V22), enriched BP GO terms included regulation of transcription, DNA-templated, oxidation–reduction process, and response to wounding; enriched CC GO terms included cytoplasm, vacuole, and apoplast; and enriched MF GO terms had transcription factor activity, sequence-specific DNA binding, iron ion binding, and heme binding (Figure 4B). In Cluster 1 of Analysis Group 2 (CK compared with V14 and V15), enriched BP GO terms included protein phosphorylation, defense response, response to chitin, response to wounding, and response to abacisic acid; enriched CC GO terms included plasma membrane and cytoplasm; and enriched MF GO terms had transcription factor activity, sequence-specific DNA binding, calcium ion binding, and calmodulin binding (Figure 4C). In Cluster 2 of Analysis Group 2 (CK compared with V14 and V15), enriched BP GO terms included regulation of transcription, DNA-templated, response to karrikin, and circadian rhythm; enriched CC GO terms included Golgi apparatus, apoplast, and vacuole; and enriched MF GO terms had transcription factor activity, sequence-specific DNA binding, DNA binding, and metal ion binding (Figure 4D).

To further determine the functions of the DEGs in the kinetic clusters, Kyoto Encyclopedia of Genes and Genomes (KEGG) enrichment analysis was performed, revealing that in Analysis Group 1 (V0 compared with V8, V14, V15, and V22), the DEGs in Cluster 1 were mainly enriched in the phenylpropanoid biosynthesis, carbon metabolism, starch and sucrose metabolism, and carbon fixation in photosynthetic organisms (Figure 5A). The analysis results also revealed that the DEGs in Cluster 2 were mainly enriched in plant hormone signal transduction, alpha-linolenic acid metabolism, and linolenic acid metabolism (Figure 5B). In addition, the KEGG analysis showed that in Analysis Group 2 (CK compared with V14, and V15), the DEGs in Cluster 1 were mainly enriched in plant pathogen interaction and plant hormone signal transduction (Figure 5C). Moreover, the DEGs in Cluster 2 were mainly enriched in carbon metabolism, starch and sucrose metabolism, phenylpropanoid biosynthesis, and glycolysis/gluconeogenesis (Figure 5D).

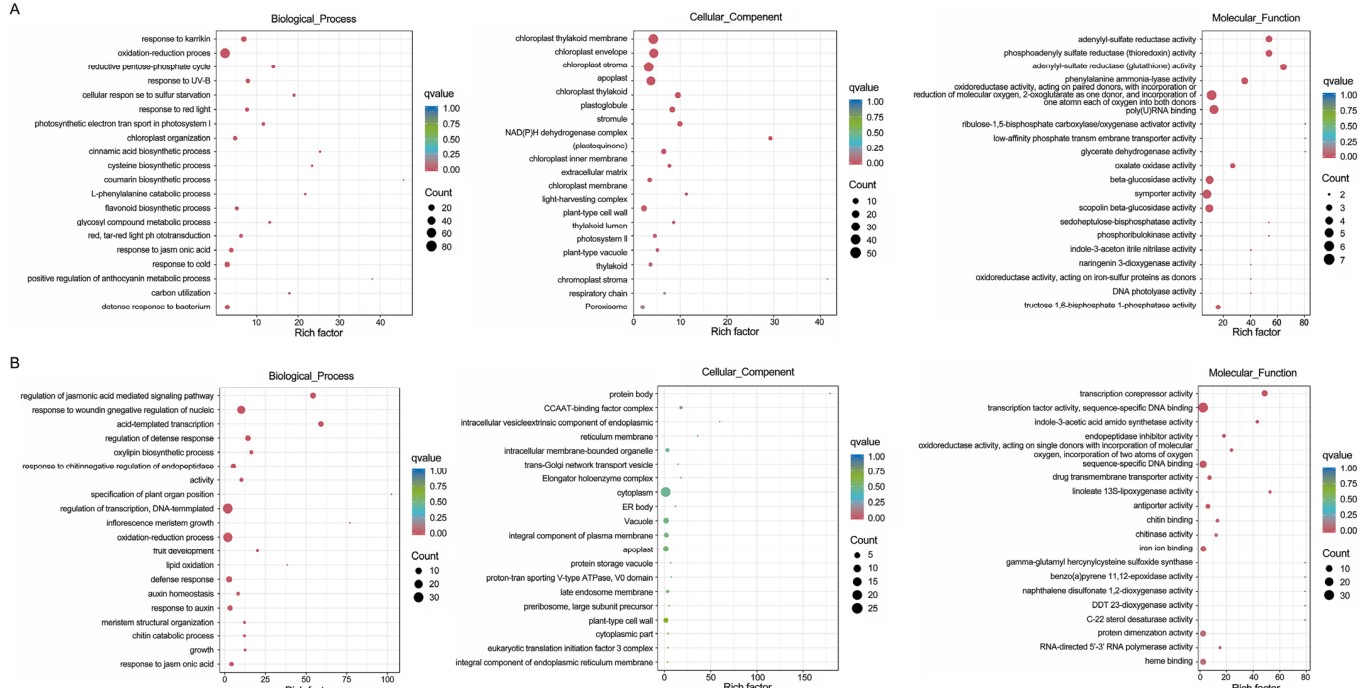

**Figure 4.** *Cont.*

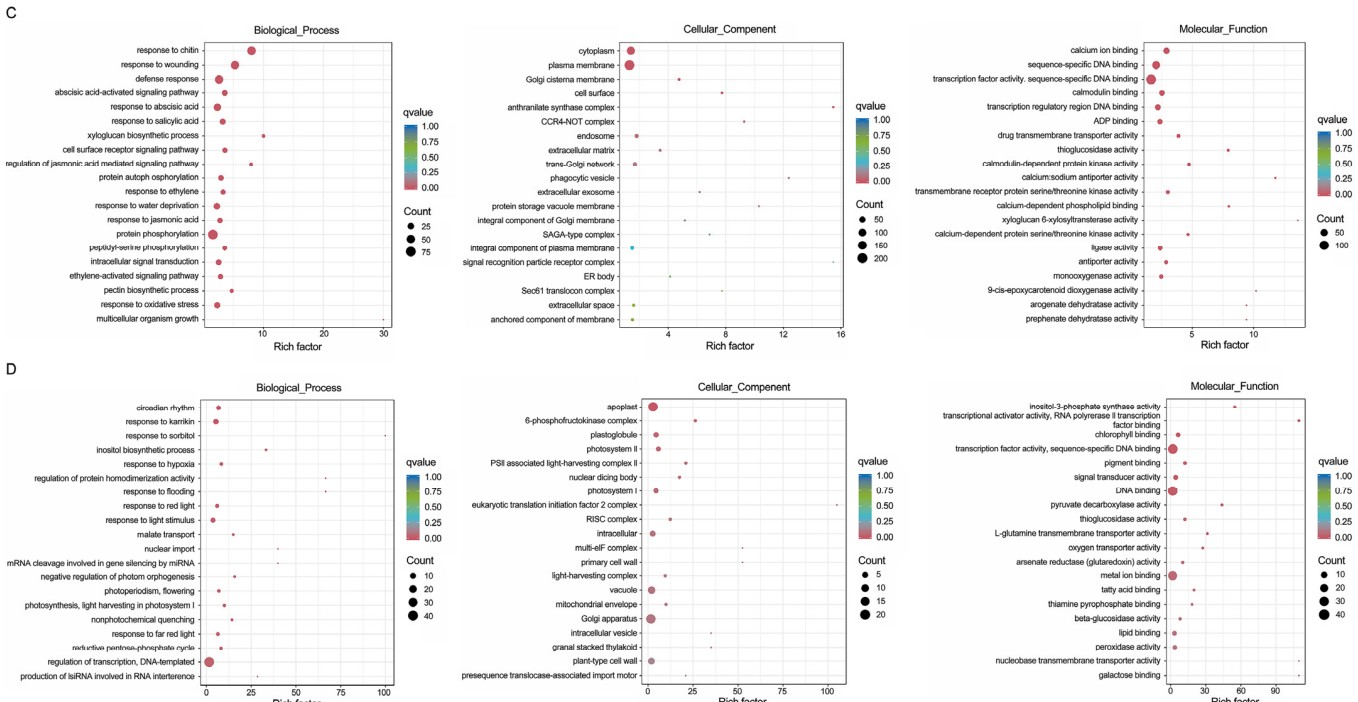

**Figure 4.** GO enrichment analysis of DEGs with transcriptional changes in the two analysis groups. (**A**) GO significant enrichment analysis of DEGs in Cluster 1 in Analysis Group 1 (V0 compared with V8, V14, V15, and V22). (**B**) GO significant enrichment analysis of DEGs in Cluster 2 in Analysis Group 1 (V0 compared with V8, V14, V15, and V22). (**C**) GO significant enrichment analysis of DEGs in Cluster 1 in Analysis Group 2 (CK compared with V14 and V15). (**D**) GO significant enrichment analysis of DEGs in Cluster 2 in Analysis Group 2 (CK compared with V14 and V15).

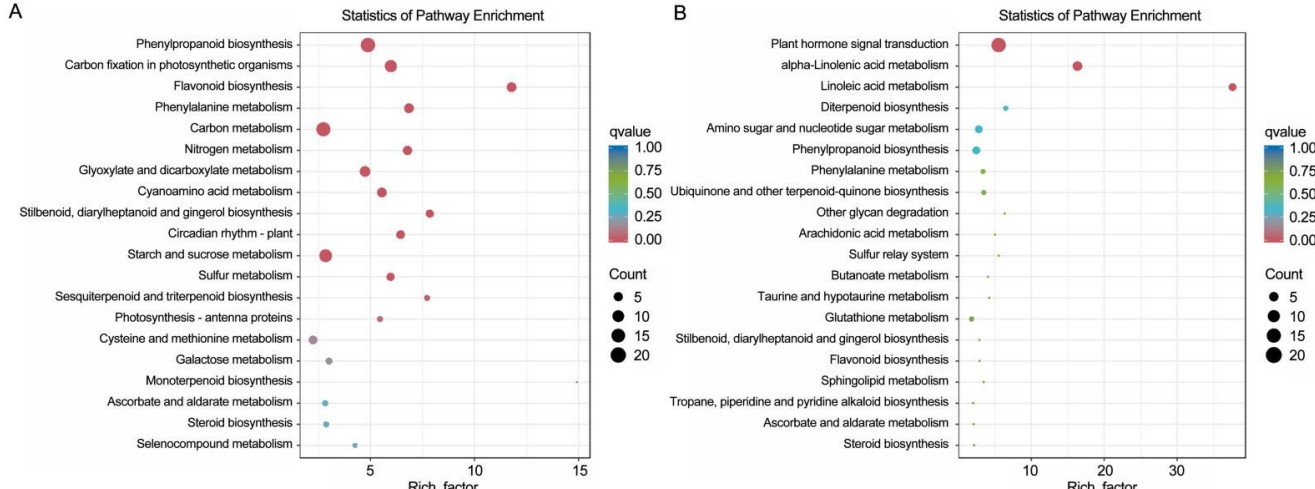

**Figure 5.** *Cont*.

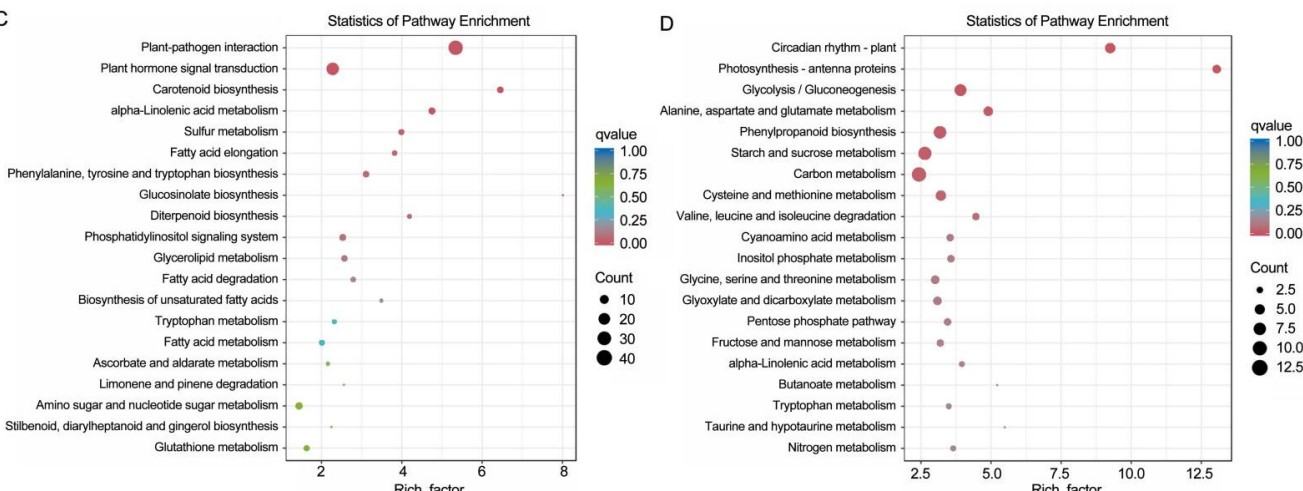

**Figure 5.** The KEGG enriched analysis of the DEGs in the comparison groups. (**A**) KEGG pathway enrichment analysis of DEGs in Cluster 1 in Analysis Group 1 (V0 compared with V8, V14, V15, and V22). (**B**) KEGG pathway enrichment analysis of DEGs in Cluster 2 in Analysis Group 1 (V0 compared with V8, V14, V15, and V22). (**C**) KEGG pathway enrichment analysis of DEGs in Cluster 1 in Analysis Group 2 (CK compared with V14 and V15). (**D**) KEGG pathway enrichment analysis of DEGs in Cluster 2 in Analysis Group 2 (CK compared with V14 and V15).

*3.5. Identification of DEGs Involved in the Plant Hormone Signaling Pathways*

Plant endogenous hormones are involved in the whole plant life process, and the effects of these hormones (gibberellins, abscisic acid, auxin, cytokinins, salicylic acid, jasmonic acid, ethylene, and brassinolide) on plant FBD have been reported [38,39]. Therefore, we screened for DEGs that may be involved in the plant hormone signaling pathways from the two clusters of coexpressed genes at the critical stage of broccoli FBD. In this study, 14 DEGs were identified as being involved in five plant hormone signaling pathways that may be associated with FBD. All DEGs of the auxin (Aux), ABA, and JA signaling pathways were downregulated under treatment at relatively low temperatures (Figure 6 and Table S6). There were more DEGs in the Aux signaling pathway than in the other two pathways, and there was only one DEG in each of the ABA and JA signaling pathways, namely, *ABI5* (BolC4t22834H) and *JAZ* (BolC1t04587H), respectively. In the Aux signaling pathway, six *Aux/IAA* genes and four *GH3* genes were significantly differentially expressed in the broccoli shoot tips under treatment at different temperatures, and the expression levels of BolC1t04847H, BolC1t04849H, BolC1t04852H, and BolC1t00171H showed five- to fifteen-fold changes. In the ETH and SA signaling pathways, DEGs were upregulated under treatment at relatively low temperatures, and the expression of BolC3t17895H increased with increasing treatment time (Figure 6 and Table S6).

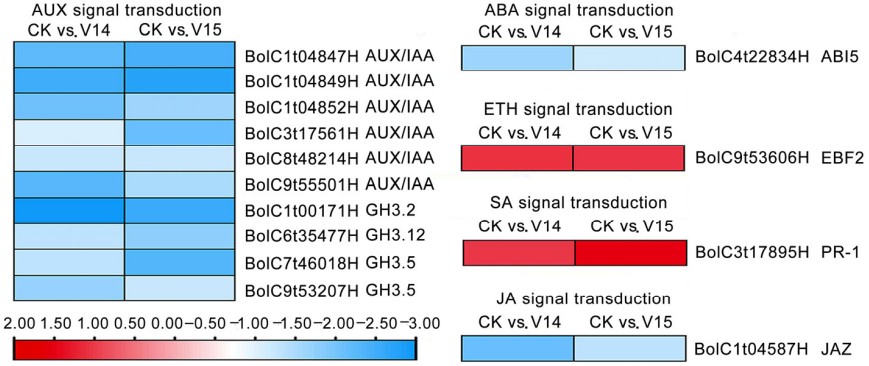

**Figure 6.** Heatmaps of phytohormone signaling genes in CK compared with V14 and V15. Red rectangles represent the upregulated genes, and blue rectangles represent the downregulated genes.

All information for each gene list can be found in Table S6. CK represents the samples from the plants under 25 °C treatment for 15 d. V14 and V15 represent the samples from the plants under 17 °C treatment for 14 d and 15 d, respectively. The numbers on the color key strip represent Log2ration.

### 3.6. Identification of Differentially Expressed TFs (DETFs)

TFs are one of the critical factors regulating gene expression at the transcriptional level [40]. The reproductive development of plants is dominated by complex and intricate gene regulatory networks of TFs [41]. Herein, there were 43 DETFs in the two clusters of coexpressed genes at the critical stage of broccoli FBD, which were grouped into nine families (Figure 7, Table S7). In the AP2/ERFBP TF family, only one (BolC4t28768H, *ERF071*) out of fifteen genes was upregulated, while the expression of the other genes was downregulated under treatment at relatively low temperatures. Among these genes, the expression of BolC3t17033H (*ERF4L*), BolC3t21433H (*ERF8*), BolC5t31450 (*ERF11L*), and BolC6t40184H (*ERF013L*) changed more than five-fold in CK compared with V14 (Figure 7, Table S7).

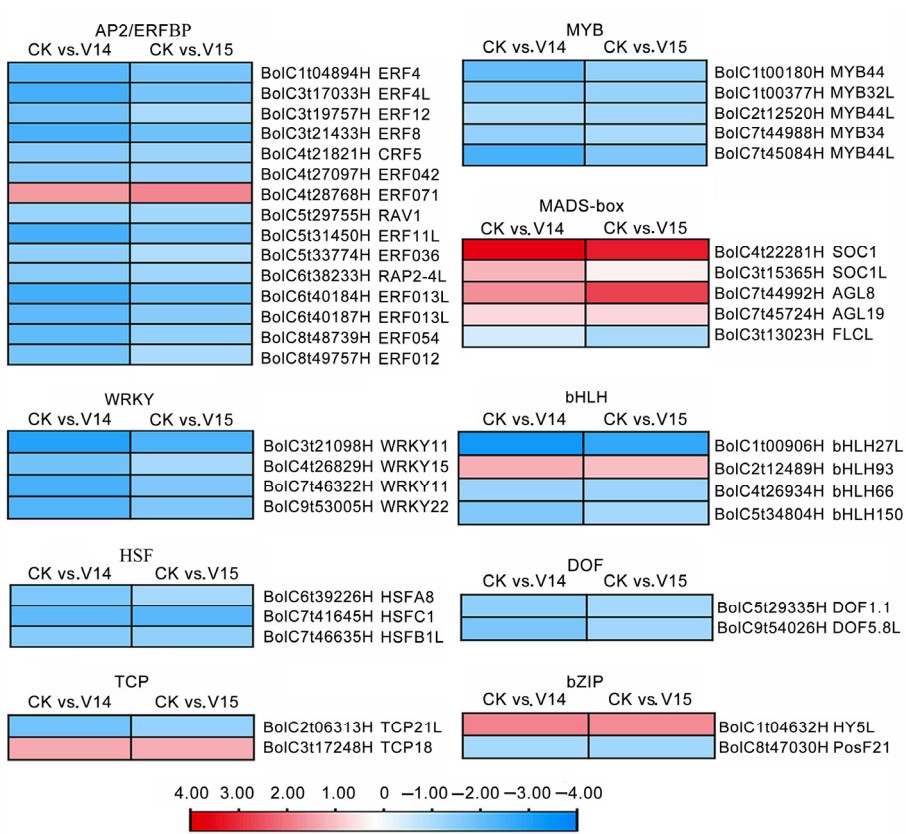

**Figure 7.** Heatmaps of DETFs in CK compared with V14 and V15. Red rectangles represent the upregulated genes, and blue rectangles represent the downregulated genes. All information for each gene list can be found in Table S7. CK represents the samples from the plants under 25 °C treatment for 15 d. V14 and V15 represent the samples from the plants under 17 °C treatment for 14 d and 15 d, respectively. The numbers on the color key strip represent Log2ration.

All DEGs of the MYB, WRKY, HSF, and DOF TF families in the broccoli shoot tips under treatment at different temperatures were downregulated. The expression of BolC3t21098H (*WRKY11*) changed 8.7-fold in CK compared with V14, and the expression of BolC7t46322H (*WRKY15*) and BolC7t45084H (*MYB44L*) changed more than 5-fold in CK compared with V14 (Figure 7, Table S7).

The MADS-box gene family plays a crucial role in flower and fruit development, and its role in the differentiation of the floral meristem into various floral organs has been widely explored [42]. In this study, four genes in the MADS-box TF family were upregulated, and one gene was downregulated under treatment at relatively low temperatures (Figure 7,

Table S7). The expression of BolC4t22281 (*SOC1*) showed more than 10-fold changes under treatment at relatively low temperatures. The expression of BolC7t44992H (*AGL8*) increased with the relatively low-temperature treatment time, and the expression level of V15 was twice that of V14 (Figure 7, Table S7).

In this study, among four bHLH TFs, BolC1t00906H (*bHLH27L*), BolC4t26934H (*bHLH66*), and BolC5t34804 (*bHLH150L*) displayed downregulated expression levels. At the same time, BolC2t12489H (*bHLH93*) showed upregulated expression level under treatment at relatively low temperatures, where the expression of the upregulated gene *bHLH27L* showed more than six-fold changes in both groups compared (Figure 7, Table S7).

Within the TCP and bZIP transcription factor families, one upregulated gene and one downregulated gene were screened. The DEGs of the TCP transcription factor were BolC3t17248H (*TCP18*) and BolC2t06313H (*TCP21L*), and the DEGs of the bZIP transcription factor were BolC1t04632H (*HY5L*) and BolC8t47030H (*PosF21*) (Figure 7, Table S7).

### 3.7. Identification of DEGs Involved in the FBD Pathways

Six flower development signaling pathways have been described, including the photoperiod pathway, vernalization pathway, autonomic pathway, GA pathway, temperature sensitivity pathway, and age pathway [14]. Herein, based on *Arabidopsis thaliana* research, GO annotation, and KEGG enrichment analysis of the two clusters of coexpressed genes at the critical stage of broccoli FBD, 16 DEGs associated with flower development pathways were filtered. Among them, nine genes were upregulated and seven genes were downregulated (Figure 8, Table S8). Many MADS-box genes play significant roles in plant flower development. In our study, the expression levels of the MADS-box genes BolC3t15365H (*SOC1*), BolC4t22281H (*SOC1*), BolC7t45724H (*AGL19*), and BolC7t44992H (*AGL8*) were upregulated under treatment at relatively low temperatures.

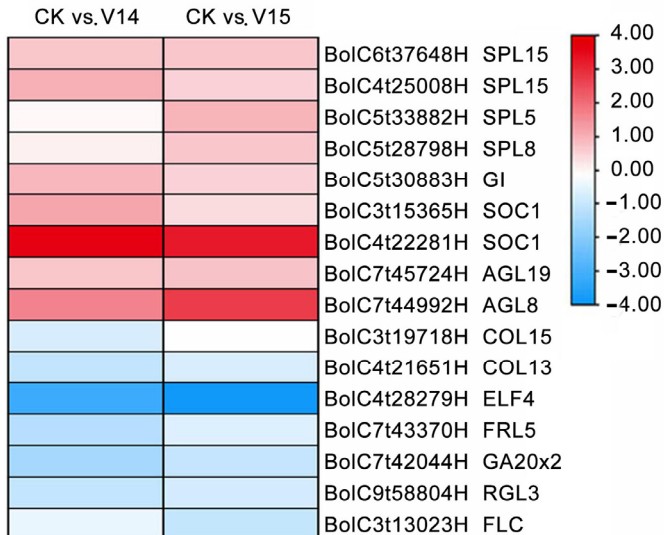

**Figure 8.** Heatmaps of genes related to flower formation in CK compared with V14 and V15. Red rectangles represent the upregulated genes, and blue rectangles represent the downregulated genes. All information for each gene list can be found in Table S8. CK represents the samples from the plants under 25 °C treatment for 15 d. V14 and V15 represent the samples from the plants under 17 °C treatment for 14 d and 15 d, respectively. The numbers on the color key strip represent Log2ration.

Previous studies have shown that *FLC* is an important gene in the vernalization pathway and the autonomous pathway, while *FRL* is an *FLC* promoter FRIGIDA complex (FRI-C) [43]. The expression levels of BolC3t13023H (*FLC*) and BolC7t43370H (*FRL*) were downregulated under treatment at relatively low temperatures.

GI and ELF4 regulate plant flowering by regulating the circadian rhythm of the biological clock in the photoperiodic pathway [44,45]. *CONSTANS-LIKE* (*COL*) genes are

important signaling components in the photoperiod pathway and flowering regulation pathway [46]. In this research, the expression of BolC5t30883H (*GI*) was upregulated, while the expression of BolC4t28279H (*ELF4*), BolC3t19718H (*COL15*), and BolC4t21651H (*COL13*) were downregulated under treatment at relatively low temperatures.

*GA2ox2* is a gene in the GA pathway regulating GA biosynthesis, and *RGL3* is an important gene in the GA signaling pathway [14]. The expression of BolC7t42044H (*GA2ox2*) and BolC9t58804H (*RGL3*) were downregulated under treatment at relatively low temperatures.

Temperature and age pathways promote flower formation by regulating *SPL* expression in *Arabidopsis thaliana*. In this research, the expression levels of BolC6t37648H (*SPL15*), BolC4t25008H (*SPL15*), BolC5t33882H (*SPL5*), and BolC5t28798H (*SPL8*) were upregulated under treatment at relatively low temperatures.

### 3.8. Analysis of Enrichment with Binding Sites for DETFs Involved in the FBD Pathway

Although 564 *Arabidopsis thaliana* TFs motifs entered the motif enrichment analysis, the results of overrepresented motifs for DETF genes (Figures 6–8) in the promoters of DEGs from CK compared with V14 and V15 are shown in Figure 9. The top four TF classes were basic leucine zipper factors (bZIP), GCM domain factors, MADS-box factors, and C2H2 zinc finger factors, indicating that their closely related TFs may be directly involved in the transcription regulation of broccoli FBD.

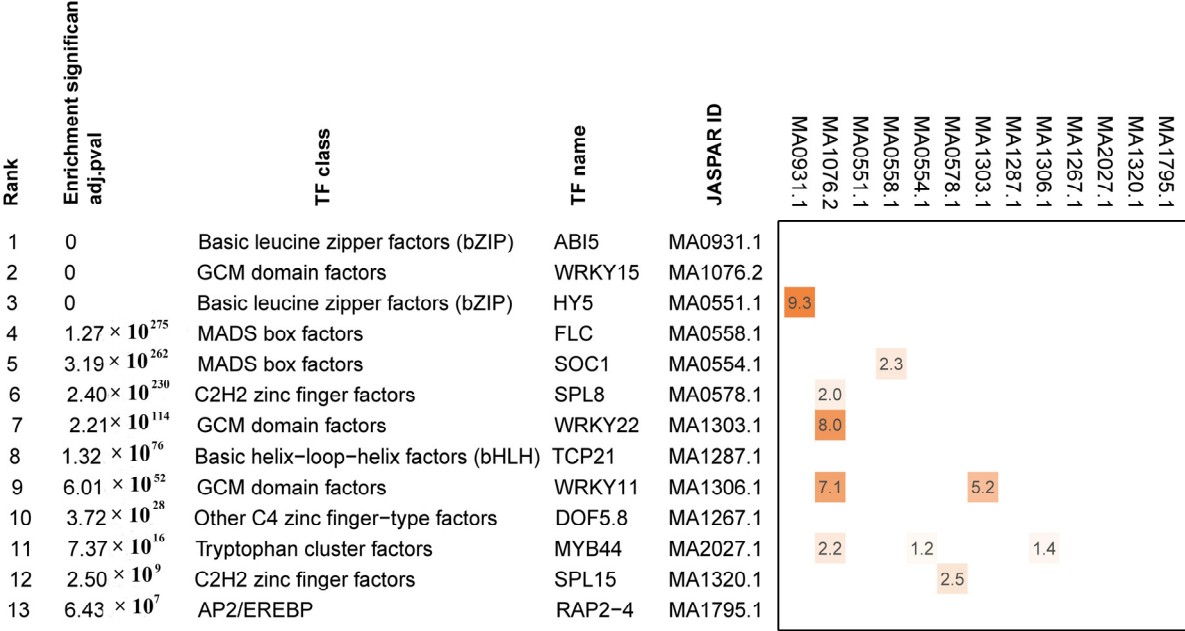

**Figure 9.** Ranking and classification of the overrepresented motifs for DETF genes (Figures 6–8) in the promoters of DEGs from CK compared with V14 and V15. Row names indicated the ranks of motifs, the significance of their enrichment computed by ESDEG [(log10(Padj))], the respective TF class from JASPAR, the names of TFs, and the JASPAR ID. The matrix heatmap indicates pairwise similarities of motifs as estimated by the permutation test using the ESDEG tools [https://github.com/ubercomrade/esdeg (accessed on 16 November 2023)]. Empty cells imply significantly distinct motifs (*p*-value > 0.05), the shades from light orange to dark orange denote significantly similar motifs (*p*-value < 0.05), and the numbers indicate the logarithm of pairwise significance: $\log_{10}(p\text{-value})$.

### 3.9. qRT-PCR Analysis Results

To validate the reliability of the RNA-seq results, ten DEGs (four genes related to plant hormone signaling, four TFs, and two genes related to the FBD pathways) were selected for qRT-PCR validation analysis in six treatments (V0, V8, V14, V15, V22, and CK) (Figure 10). Comparing the results of RNA-seq with qRT-PCR, it was found that the expression trends

of ten DEGs at different temperatures and different treatment times at the same temperature were compatible with the sequencing results, suggesting that the RNA-seq results were valid and reliable. Among the ten DEGs, BolC7t44992H (*AGL8*), related to FBD, was gradually upregulated with prolonged treatment at relatively low temperatures.

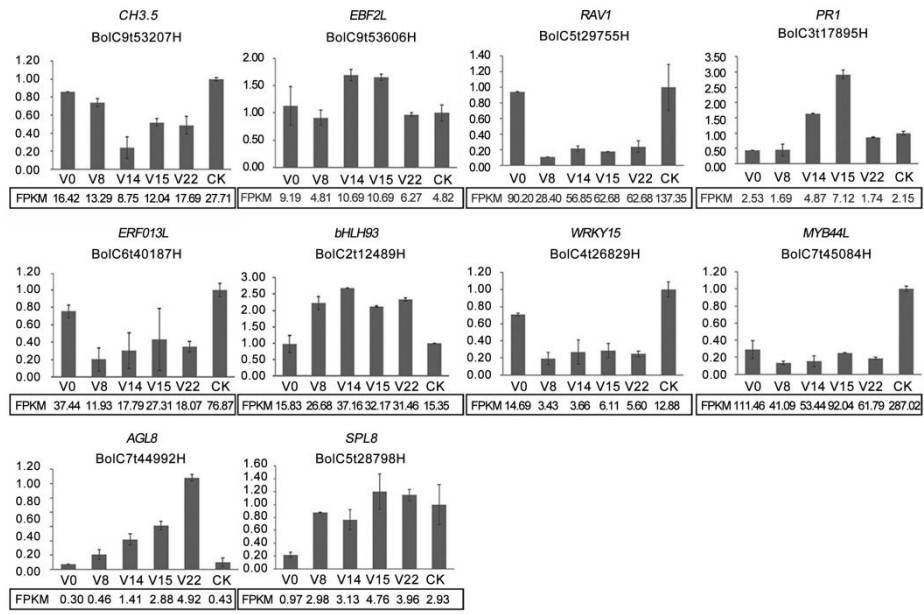

**Figure 10.** qRT-PCR verification of RNA-seq results based on gene expression level. FPKM: fragments per kilobase of transcript per million fragments mapped. Each bar represents the average plus the standard deviation of three biological replicates.

## 4. Discussion

Low temperature is necessary for FBD of Arabidopsis late-flowering ecotypes, Chinese cabbage, and radish [47–49]. Similarly, broccoli plants require exposure to low temperatures before FBD and flower formation; so, they usually grow in areas with an average temperature below 18 °C. In this research, the changes in the broccoli shoot tips were observed under a stereomicroscope and by paraffin-sectioning micrographs. FBD began at 15 days in the treatment group (17 °C/9 °C). However, the broccoli shoot tips remained in vegetative growth in the control group (25 °C/17 °C). In addition, transcriptome information was gained by high-throughput sequencing of broccoli shoot tips grown under different temperatures and treatment times. Differences in expression patterns among samples may provide a basis for identifying important genes and pathways during FBD. Coexpression analysis displayed 867 DEGs in the comparison of V0, V8, V14, V15, and V22 and 2059 DEGs in the comparison of CK, V14, and V15. GO and KEGG enrichment analyses were performed for these DEGs. Then, we further analyzed the significant enrichment of DEGs between the critical stages of FBD (V14, V15) and the control treatment (CK), including genes related to plant hormone signaling pathways (Figure 6), transcription factors (Figure 7), and flower development pathways (Figure 8). It has been reported that under specific conditions, regulation of hormone signaling is often achieved by the pooling of different hormone signals and then altering the expression levels of the key flower-forming genes [50]. It has also been shown that TFs interact with genes related to the flower development pathways to regulate plant flowering [51,52]. Meanwhile, analysis of enrichment with binding sites for DETFs involved in the FBD pathway indicated that the top four TF classes were bZIP, GCM domain factors, MADS-box factors, and C2H2 zinc finger factors, whose closely related TFs were directly involved in the transcription regulation of broccoli FBD.

Proteins with the basic leucine zipper motif (bZIP) domains are present in all eukaryotes, and bZIP TFs are located in the nucleolus and participate in plant growth, floral development, and other biological processes [53]. In this study, there were three genes

[BolC4t22834H (*ABI5*), BolC1t04632H (*HY5L*), and BolC8t47030H (*PosF21*)] belonging to the bZIP TF class. *ABI5* transcription factor regulates floral transition in *Arabidopsis thaliana* through the central mediator *FLC* [54]. In our study, *ABI5* (BolC4t22834H) was downregulated at relatively low temperatures, suggesting that *ABI5* may play a negative regulatory role during broccoli FBD. This is compatible with previous studies on *Arabidopsis thaliana* [55] and saffron crocus [56]. In addition, the expression level of BolC1t04632H (*HY5L*) was upregulated. At the same time, that of BolC8t47030H (*PosF21*) was downregulated under treatment at relatively low temperatures, suggesting that the upregulated expression of BolC1t04632H (*HY5L*) and downregulated expression of BolC8t47030H (*PosF21*) may promote broccoli FBD under treatment at relatively low temperatures.

WRKY TFs belonging to GCM domain factors class participate in organelle composition, dwarfism, flowering, fruiting, dormancy, and senescence of plant [51,57–59]. Under normal conditions, *Arabidopsis thaliana* plants overexpressing *CsWRKY7* showed a late-flowering phenotype [60]. Under short-day (SD) conditions, WRKY13 interacts with SPL10 to repress the transcriptional function of SPL10, thereby inhibiting flowering [51]. Strawberry *FvWRKY71* accelerates flowering by directly regulating the expression of *FvFUL*, *FvSEP1*, *FvAGL42*, *FvLFY*, and *FvFPF1* [52]. In this research, the expression levels of four *WRKY* TFs [BolC3t21098H (*WRKY11*), BolC4t26829H (*WRKY15*), BolC7t46322H (*WRKY11*), and BolC9t53005H (*WRKY22*)] were downregulated under treatment at relatively low temperatures, indicating that *WRKYs* may inhibit broccoli FBD under treatment at relatively low temperatures.

The role of the MADS-box gene family in the floral meristem differentiation into various floral organs has been widely explored [42]. MADS-box genes including *SOC1*, *FLC*, and *AGLs* participated in the regulation of flowering timing in *Arabidopsis thaliana* [61–64]. *SOC1* and *AGLs* induced the flowering of *Arabidopsis thaliana*, while an opposite effect on flowering was exerted by *FLC* [18,65,66]. In this research, four genes in the MADS-box TF family were upregulated, and one gene was downregulated under treatment at relatively low temperatures, suggesting that the upregulated expression of BolC4t22281 (*SOC1*), BolC3t15365H (*SOC1-like*), BolC7t44992H (*AGL8*), and BolC7t45724H (*AGL19*) and the downregulated expression of BolC3t13023H (*FLC*) promote broccoli FBD under treatment at relatively low temperatures.

Temperature and age pathways promote flower formation by regulating *SPL* expression in *Arabidopsis thaliana*. The *SPL* protein family belongs to the C2H2 zinc finger factor class and contains the highly conserved SBP domain [57], which binds to the promoter region of downstream target genes and regulates their expression. *SPL* promotes the transcription of *FT*, *SOC1*, *AGL24*, and other genes [67–69]. The FT/SOC1-mediated photoperiod pathway in *Arabidopsis thaliana* promotes flowering by directly binding to the promoters of the genes and inducing the *SPL3*, *SPL4*, and *SPL5* genes [68,70]. *SPL8* regulates flowering by positively regulating GA signaling during flower development in *Arabidopsis thaliana* [67]. Two closely related genes, *SPL9* and *SPL15*, represent an evolutionary branch of the *Arabidopsis thaliana* SPL family with important functions in flower-forming induction and floral organ development [71,72]. *SPL15* directly activates the transcription of the floral regulators *FUL* and *miR172b* in SAM during floral induction, promoting flowering [73]. Overexpression of the *Lilium LbrSPL15* gene in *Arabidopsis thaliana* resulted in early flowering in transgenic plants [74]. In the present study, the expression levels of BolC6t37648H (*SPL15*), BolC4t25008H (*SPL15*), BolC5t33882H (*SPL5*), and BolC5t28798H (*SPL8*) were upregulated under treatment at relatively low temperatures, indicating that they may promote broccoli FBD. This is compatible with previous studies of *Arabidopsis thaliana* [67,68,73].

## 5. Conclusions

First, after 15 days of low-temperature treatment, broccoli flower buds began to differentiate. However, the control (normal temperature conditions) was still in the vegetative growth stage, indicating that relatively low temperatures successfully induced flower bud formation. Second, DEGs related to Aux, ABA, ETH, SA, and JA signaling pathways showed different expression patterns under different temperature treatments, which indicated that there may be multiple hormonal interactions in broccoli to regulate FBD induced by relative

low temperatures. In addition, the auxin signaling pathway plays a key role in regulating the broccoli FBD induced by relatively low temperatures. Furthermore, bZIP, GCM domain factors, MADS-box factors, and C2H2 zinc finger factors possess enriched motifs, indicating that their closely related DETFs *ABI5*, *HY5L*, *WRKY11*, *WRKY15*, *WRKY22*, *SOC1*, *AGL8*, *FLC*, *SPL8*, and *SPL15* may be directly involved in the transcription regulation of broccoli FBD.

**Supplementary Materials:** The following supporting information can be downloaded at: https://www.mdpi.com/article/10.3390/horticulturae9121353/s1, Figure S1: Correlation analysis between samples; Table S1: Details of primers for qRT-PCR in this study; Table S2: Sequencing statistics and comparison of each library; Table S3: Statistical table of DEGs between V0 and V8, between V0 and V14, between V0 and V15, between V0 and V22, between CK and V14, and between CK and V15; Table S4: Co-expressed genes in broccoli shoot tips under different durations of treatment at relatively low temperatures; Table S5: Co-expressed genes at the critical stage of flower bud differentiation in broccoli; Table S6: DEGs at the critical stage of flower bud differentiation in broccoli involved in the plant hormone signaling pathway; Table S7: DETFs at the critical stage of flower bud differentiation in broccoli; Table S8: DEGs involved in the flower bud differentiation pathway of broccoli.

**Author Contributions:** Conceptualization, J.X., L.H., M.L. and J.Z.; investigation, W.C., X.H., Z.Z., B.W., Y.J., R.B. and X.Z.; data curation, X.H., Y.J. and J.Z.; writing—original draft preparation, W.C., X.H. and B.W.; writing—review and editing, J.Z.; visualization, X.H. and Z.Z.; project administration, J.Z.; funding acquisition, W.C. and J.Z. All authors have read and agreed to the published version of the manuscript.

**Funding:** This work was supported by the Natural Science Foundation of Shanxi Province (Grant No. 202203021211280 and 201701D221188) and the Special Plan Project of Shanxi Province for the Transformation of Patent (Grant No. 202201017).

**Data Availability Statement:** The authors confirm that the data supporting the findings of this study are available within NCBI (PRJNA1022574).

**Conflicts of Interest:** The authors declare no conflict of interest.

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
