# Peer review of "The Molecular Mechanism of Relatively Low-Temperature-Induced Broccoli Flower Bud Differentiation Revealed by Transcriptomic Profiling"

_horticulturae, doi:10.3390/horticulturae9121353_

Round 1
Reviewer 1 Report (Previous Reviewer 2)
Comments and Suggestions for Authors
I can'tWhe to reject the novelty of results. RNA-seq, though is not cheap, is still a routine approach. But, there are not yet many papers that try to infer TFs that regulate gene transcription almost directly from RNA-seq data. So the manuscript should move towards acceptance. The only obstacles are the technical design of the paper, correct http links, and neatness in details, and overall accuracy of data presentation in Figures as main reports of the manuscript.
Major
756
Data availability statement: The authors confirm that the data supporting the findings of this study 756 are available within NCBI (PRJNA1022574).
This link is dead too
Fig. 9 ESDEG
1. 1. Anyone can create a github site, but a proper github site contains software used in at least one paper; bad software is definitely not cited anywhere, so your citation of the ESDEG is missed, please provide the reference where it was first used previously, this is ref #1 from the github site
2. The Enrichment of certain motif does not mean that exactly this TF is important; rather it means that it or its closely related TFs are directly involved in transcription regulation. Close homologous TFs refer to the same classes or even families, that is the reason why you see class names in the Figure, and, moreover, you can also see the matrix of motifs similarity. You have RNA-seq data, you may propose your important TFs.
3. From Figure caption ...The results of overrepresented motifs for the selected TFs… How many motifs (TFs) entered the motif enrichment analysis? Figure shows all enriched motifs?
AGL8 is absent in JASPAR, but several its close relatives are present there (see all members of MADS box)
HY5 and ABI5 are close homologous TFs (according to the similarity of their motifs, see ESDEG Figure), this Figure tells that the most important TFs are from bZIP, GSM domain, MADS box classes. The next ones are C2H2 ZF and bHLH.. and these are almost all. Your RNA-seq data actually lists the TF names, most probably AGL8 is represented by FLC & SOC1 motif too
As it was written in JASPAR 2024 paper … TFs can be classified into structurally related families based on their DBDs. TFs with DBDs from the same structural family tend to recognize similar DNA sequence motifs, except for zinc finger proteins …
ESDEG strongly supports your results on bZIP and MADS-box TFs, you should briefly discuss it. Both TFs listed in your Abstract (SOC1 & AGL8) are reflected in top5 motifs in the ESDEG output, see JASPAR logo https://jaspar.elixir.no/matrix-clusters/plants/, may go there http://cisbp.ccbr.utoronto.ca/index.php with query AGL*
So the simple questions to answer in discussion – which DE TF genes possess enriched motifs, you may list them even for two three classes as in Abstract you wrote only about MADS box TFs as the most important, just overlap your Figure ESDEG & Table S7.
Minor yet important
141
Where to find GCA_900416815.2 ?
E.g. https://ngdc.cncb.ac.cn/gwh/Genome/48/show
280
To reveal the biological processes (BP), cellular component (CC), biological processes 280 (BP), and molecular function (MF) ->
To reveal the biological processes (BP), cellular components (CC), and molecular functions (MF)
In general, English still needs to be scrutinized thoroughly, use at least free online translate tools
329
Identification of DEGs involved in the plant hormone signaling pathway -> Identification of DEGs involved in the plant hormone signaling pathways
The subsequent text directly reports about several hormones and several pathways.
333
pathway -> pathways
433
ELF4 is not TF, see http://planttfdb.gao-lab.org/download.php, see also your section 4.3 DEGs involved in the FBD pathway, where ELF4 – TFs are in 4.2 DETFs involved in broccoli FBD
Figures 6, 7, 8. Explain the meaning of colors, e.g. numbers above a color key strip, p-value, fold or anything else?
356
TF is one of the key factors -> TFs are one of the key factors
… You as many other study many of them at once
384
The a/an article is incorrect if you are pointing to something specific
One upregulated and one downregulated gene in each of the TCP and bZIP transcription factor families was detected.
433, 450
Two fig. 9
provide distinct numbers for distinct figures
Various piece of text mention the same terms (bHLH, C2H2 zinc fingers, MADS box etc) as classes or families
You on the one page mention the bHLH is family and on the another page it is already bHLH class. It is misleading. Several levels of hierarchy as in JASPAR/TFClass/PlantTFclass is better approach than just a one level of families which integrate anything without taking care of its hierarchical level. There are many families within the bHLH class. The term class is slightly better than a superfamily (Zhang, C., Feng, R., Ma, R., Shen, Z., Cai, Z., Song, Z., Peng, B., & Yu, M. (2018). Genome-wide analysis of basic helix-loop-helix superfamily members in peach. PloS one, 13(4), e0195974. https://doi.org/10.1371/journal.pone.0195974)
514
ABI5 transcription factors regulate floral transition in Arabidopsis thaliana through
->
ABI5 transcription factor regulates floral transition in Arabidopsis thaliana through
Comments on the Quality of English Language
English may be substantially improved, it is neither brilliant not very awfulI
Author Response
We have revised the manuscript after careful consideration of your comments. Here are our responses in detail. The revised parts used Microsoft Word's built-in track changes function to highlight any changes.
1. Data availability statement: The authors confirm that the data supporting the findings of thisstudy 756 are available within NCBI (PRJNA1022574). This link is dead too.
Response: We have published the RNA-seq data in NCBI (PRJNA1022574). The link is https://www.ncbi.nlm.nih.gov/search/all/?term=PRJNA1022574.
2. Fig. 9 ESDEG
(1) Anyone can create a github site, but a proper github site contains softwareused in at least one paper; bad software is definitely not cited anywhere, so your citation of the ESDEG is missed, please provide the reference where it was first used previously, this is ref #1 from the github site.
Response: We have added references in section 2.5.
Original manuscript
Motif overrepresentation of the promoters of DEGs from CK compared with V14 and V15 were estimated by the ESDEG tools (https://github.com/ubercomrade/esdeg).
Revised manuscript
Motif overrepresentation of the promoters of DEGs from CK compared with V14 and V15 were estimated by the ESDEG tools (https://github.com/ubercomrade/esdeg) [36].
- Oshchepkov, D.; Chadaeva, I.; Kozhemyakina, R.; Shikhevich, S.; Sharypova, E.; Savinkova, L.; Klimova, N.V.; Tsukanov, A.; Levitsky, V.G.; Markel, A.L. Transcription factors as important regulators of changes in behavior through domestication of gray rats: quantitative data from RNA sequencing. J. Mol. Sci.2022, 23, 12269. DOI: 10.3390/ijms232012269.
(2)The Enrichment of certain motif does not mean that exactly this TF is important; rather it means that it or its closely related TFs are directly involved in transcription regulation. Close homologous TFs refer to the same classes or even families, that is the reason why you see class names in the Figure, and, moreover, you can also see the matrix of motifs similarity. You have RNA-seq data, you may propose your important TFs.
Response: We have rewritten section 3.8.
Original manuscript
The results of overrepresented motifs for the selected TFs in the promoters of DEGs from CK compared with V14 and V15 are shown in Figure 9. The top five motifs in the DEG motif enrichment list were ABI5, WRKY15, HY5, FLC, and SOC1, indicating that they play a crucial role in broccoli FBD. In addition, the ELF4 motif and AGL8 motif are not shown in Figure 9, because they do not exist in JASPAR database. However, they may also be very important in the FBD of broccoli.
Revised manuscript
Although 564 Arabidopsis thaliana TFs motifs entered the motif enrichment analysis, the results of overrepresented motifs for DETFs genes (Figures 6, 7, and 8) in the promoters of DEGs from CK compared with V14, and V15 were shown in Figure 9. The top four TF classes were basic leucine zipper factors (bZIP), GCM domain factors, MADS box factors, and C2H2 zinc finger factors, indicating that their closely related TFs may be directly involved in transcription regulation of broccoli FBD.
(3) From Figure caption ...The results of overrepresented motifs for the selected TFs… How many motifs (TFs) entered the motif enrichment analysis? Figure shows all enriched motifs?
Response: 564 Arabidopsis thaliana motifs (TFs) entered the motif enrichment analysis. Figure 9 showed motifs of DETFs genes (Figure 6, 7, and 8) are enriched in promoters of DEGs from CK compared with V14 and V15.
(4) AGL8 is absent in JASPAR, but several its close relatives are present there (see all members of MADS box)
HY5 and ABI5 are close homologous TFs (according to the similarity of their motifs, see ESDEG Figure), this Figure tells that the most important TFs are from bZIP, GSM domain, MADS box classes. The next ones are C2H2 ZF and bHLH.. and these are almost all. Your RNA-seq data actually lists the TF names, most probably AGL8 is represented by FLC & SOC1 motif too
As it was written in JASPAR 2024 paper … TFs can be classified into structurally related families based on their DBDs. TFs with DBDs from the same structural family tend to recognize similar DNA sequence motifs, except for zinc finger proteins …
ESDEG strongly supports your results on bZIP and MADS-box TFs, you should briefly discuss it. Both TFs listed in your Abstract (SOC1 & AGL8) are reflected in top5 motifs in the ESDEG output, see JASPAR logo https://jaspar.elixir.no/matrix-clusters/plants/, may go there http://cisbp.ccbr.utoronto.ca/index.php with query AGL*
So the simple questions to answer in discussion – which DE TF genes possess enriched motifs, you may list them even for two three classes as in Abstract you wrote only about MADS box TFs as the most important, just overlap your Figure ESDEG & Table S7.
Response: We have rewritten the discussion after carefully considering your comments.
3. 141
Where to find GCA_900416815.2 ?
E.g. https://ngdc.cncb.ac.cn/gwh/Genome/48/show
Response: The 141 lines of the manuscript clearly show the link of GCA_900416815.2 (https://www.ncbi.nlm.nih.gov/assembly/GCA_900416815.2).
4. 280
To reveal the biological processes (BP), cellular component (CC), biological processes (BP), and molecular function (MF) ->
To reveal the biological processes (BP), cellular components (CC), and molecular functions (MF)
In general, English still needs to be scrutinized thoroughly, use at least free online translate tools.
5. 329
Identification of DEGs involved in the plant hormone signaling pathway -> Identification of DEGs involved in the plant hormone signaling pathways
The subsequent text directly reports about several hormones and several pathways.
6. 333
pathway -> pathways
7. 356
TF is one of the key factors -> TFs are one of the key factors
… You as many other study many of them at once
8. 384
The a/an article is incorrect if you are pointing to something specific
One upregulated and one downregulated gene in each of the TCP and bZIP transcription factor families was detected.
9. 514
ABI5 transcription factors regulate floral transition in Arabidopsis thaliana through
->
ABI5 transcription factor regulates floral transition in Arabidopsis thaliana through
Response: We carefully checked and revised the spelling of the words and grammar in the whole manuscript.
10. 433
ELF4 is not TF, see http://planttfdb.gao-lab.org/download.php, see also your section 4.3 DEGs involved in the FBD pathway, where ELF4 – TFs are in 4.2 DETFs involved in broccoli FBD
Response: We have refined the discussion section, and the content you mentioned has been removed.
11. Figures 6, 7, 8. Explain the meaning of colors, e.g. numbers above a color key strip, p-value, fold or anything else?
Response: We described the numbers on the colored bars in Figures 6, 7, 8. The numbers on the color key strip represent Log2ration.
12. 433, 450
Two fig. 9
provide distinct numbers for distinct figures
Response: The last figure is Figure 10. We have changed the last figure from Figure 9 to Figure 10.
13. Various piece of text mention the same terms (bHLH, C2H2 zinc fingers, MADS box etc) asclasses or families
You on the one page mention the bHLH is family and on the another page it is already bHLH class. It is misleading. Several levels of hierarchy as in JASPAR/TFClass/PlantTFclass is better approach than just a one level of families which integrate anything without taking care of its hierarchical level. There are many families within the bHLH class. The term class is slightly better than a superfamily (Zhang, C., Feng, R., Ma, R., Shen, Z., Cai, Z., Song, Z., Peng, B., & Yu, M. (2018). Genome-wide analysis of basic helix-loop-helix superfamily members in peach. PloS one, 13(4), e0195974. https://doi.org/10.1371/journal.pone.0195974)
Response: We carefully checked and revised the whole manuscript.
Reviewer 2 Report (New Reviewer)
Comments and Suggestions for Authors
The article explores the molecular mechanism underlying low-temperature induced flower bud differentiation (FBD) in broccoli. The study employs a comparative analysis of broccoli plants subjected to low-temperature treatment (17°C/9°C, 16 h/8 h) with control plants grown under normal temperature conditions (25°C/17°C, 16 h/8 h), confirming the successful induction of FBD by low temperatures. The global transcriptomic analysis reveals crucial insights into the differential gene expression during low-temperature induced FBD, which enhances our understanding of the molecular regulation of broccoli flower development under low temperatures. The research paper provides valuable contributions to the field in Horticultural Science, especially in agricultural practices in optimizing broccoli yield and quality. In conclusion, the paper would be sufficient to merit publication in Horticulturae, though a revision is recommended which needs to include the following points.
(1) line 131
Need a detailed description of the broccoli sampling. Please indicate how large (length from stem apex) the shoot tip was taken. Does the shoot tip contain leaf primordia?
(2) Figure 1 and related part
Do not definitively describe the details of the developmental process based on observations of only one individual; there should be variation in the timing of FBDs. Ideally, you should show a result of statistical analysis. If it is difficult, at least the authors should clearly mention that the results are based on the observations of multiple individuals.
(3) Figure 1
Figure 1 should have a larger picture.
(4) Line 265-271
I do not understand the logic or meaning of this paragraph at all.
(5) Figure 3
The article lacks detailed information on the clustering analysis. The author clusters genes based on temporal variation in expression, but I have no idea how to do this and need a clear explanation.
(6) Figure 3
It would be easier to understand if the number of genes belonging to each cluster is indicated in the figure.
(7) Figure 4
It would be better to analyze the data separately by clusters. Figure 5 shows such an analysis.
(8) Line 397
Please explain in detail how you selected the 16 genes.
(9) The discussion is too long, please shorten it. It is mostly a description of genes and pathways. If an explanation is needed, it should be in the introduction. The Discussion should include interpretation of the results, explanation of new findings, how the research results may contribute to applications, etc.
Author Response
We have revised the manuscript after careful consideration of your comments. Here are our responses in detail. The revised parts used Microsoft Word's built-in track changes function to highlight any changes.
1. line 131
Need a detailed description of the broccoli sampling. Please indicate how large (length from stem apex) the shoot tip was taken. Does the shoot tip contain leaf primordia?
Response: In section 2.3, we described the broccoli samples in detail. The highlighted yellow color indicates the content which was added.
Original manuscript
At each time point, 24 shoot tips were collected from 24 individual plants and mixed to form three biological replicates.
Revised manuscript
At each time point, 24 shoot tips with a length of about four millimeter to five millimeter and containing one or two leaf primordium were collected from 24 individual plants and mixed to form three biological replicates.
2. Figure 1 and related part
Do not definitively describe the details of the developmental process based on observations of only one individual; there should be variation in the timing of FBDs. Ideally, you should show a result of statistical analysis. If it is difficult, at least the authors should clearly mention that the results are based on the observations of multiple individuals.
Response: Lines 118 to 122 in the manuscript describe this problem. In addition, the highlighted yellow color indicates the content which was added.
The morphological changes of the broccoli shoot tips were observed by stereo microscopy (Olympus SZX16), to define the developmental stages. In the first ten days, six plants were observed every two days. After ten days of treatment, six plants were observed every day. This process was repeated three times.
3. Figure 1
Figure 1 should have a larger picture.
Response: The size of Figure 1 has been adjusted to be larger.
4. Line 265-271
I do not understand the logic or meaning of this paragraph at all.
Response: This paragraph is unnecessary, so we have deleted it in the revised manuscript.
5. Figure 3
The article lacks detailed information on the clustering analysis. The author clusters genes based on temporal variation in expression, but I have no idea how to do this and need a clear explanation.
Response: Coexpression trend analysis is an important method in gene expression analysis research, which can be used to explore genes with similar expression patterns. The essence is to classify genes with similar expression trends. The main purpose of coexpression trend analysis tools is to cluster the expression data of time-series samples or samples treated with different concentrations and analyze their expression patterns. The highlighted yellow color indicates the content which was revised.
Original manuscript
To gain insight into the dynamics of gene expression changes during FBD at relatively low temperatures, a kinetic analysis was conducted using the fragments per kilobases per million reads (FPKM) data from V0, CK, V8, V14, V15, and V22.
Revised manuscript
To gain insight into the dynamics of gene expression changes during FBD at relatively low temperatures, k-means clustering was used to analyze the fragments per kilobases per million reads (FPKM) data from V0, CK, V8, V14, V15, and V22.
6. Figure 3
It would be easier to understand if the number of genes belonging to each cluster is indicated in the figure.
Response: The number in the bottom left corner of Figure 3 represents the number of genes belonging to each cluster.
7. Figure 4
It would be better to analyze the data separately by clusters. Figure 5 shows such an analysis.
Response: Figure 4 has been changed according to your suggestion. The text related to Figure 4 has also been modified.
8. Line 397
Please explain in detail how you selected the 16 genes.
Response: In section 3.7 of the manuscript, we explain in detail how to select the16 genes. The red font indicates the content which was added.
Original manuscript
Herein, 16 DEGs associated with flower development pathways were filtered from the two clusters of coexpressed genes at the critical stage of broccoli FBD.
Revised manuscript
Herein, based on Arabidopsis thaliana research, GO annotation and KEGG enrichment analysis of the two clusters of coexpressed genes at the critical stage of broccoli FBD, 16 DEGs associated with flower development pathways were filtered.
9. The discussion is too long, please shorten it. It is mostly a description of genes and pathways. If an explanation is needed, it should be in the introduction. The Discussion should include interpretation of the results, explanation of new findings, how the research results may contribute to applications, etc.
Response: We have rewritten the discussion after carefully considering your comments.
This manuscript is a resubmission of an earlier submission. The following is a list of the peer review reports and author responses from that submission.
Round 1
Reviewer 1 Report
Comments and Suggestions for Authors
As you can see on the manuscript, I appreciate that minor interventions are necessary, as for instance, the explanation of the abbreviations written in the Abstract section.

Author Response
We have revised the manuscript after careful consideration of the comments made by the three reviewers and editor. Here are our responses in detail. The revised parts used Microsoft Word's built-in track changes function to highlight any changes.
Reviewer #1
As you can see on the manuscript, I appreciate that minor interventions are necessary, as for instance, the explanation of the abbreviations written in the Abstract section.
Response: We carefully checked and revised the Abstract section.
Line 16, a variety of Brassica oleracea has been deleted.
Line 20 and 22, Single quotation marks are not required for variety names in English, so there are no changes.
Line 22, we checked and revised the whole manuscript, especially the formatting.
Line 30, The word are has been deleted.
Line 33-37, all the abbreviations have been explained in the Abstract section.
The above changes used Microsoft Word's built-in track changes function to highlight any changes.
Reviewer 2 Report
Comments and Suggestions for Authors
The merit of the authors is the analysis of transcription factor genes among differentially expressed genes.
The manuscript tries to find among differentially expressed genes (DEGs) those directly involved in transcription regulation, namely the genes of transcription factors. Notably, the conditions of RNA-seq experiment, in particular, variation in transition to the flower bud differentiation proposes that some difference in transcription regulation is expected.
Overall, the analysis of motifs of transcription factor binding sites in DEGs can substantially support the study, in particular, its main results from Fig. 7, 8. Conventional motif enrichment analysis may be performed, but application of special tool developed for analysis of RNA-seq data would be more beneficial. In particular, I propose to test whether motifs of DE TFs genes (Figure 8) are enriched in promoters of all DEGs, see example of such analysis here
Oshchepkov et al. Transcription Factors as Important Regulators of Changes in Behavior through Domestication of Gray Rats: Quantitative Data from RNA Sequencing. Int. J. Mol. Sci. 2022, 23, 12269. https://doi.org/10.3390/ijms232012269.
In this study the tool ESDEG was specially designed to detect the most reliable significantly enriched motifs of known TFs from public library (Plant Cistrome) comparing between promoters of DEGs and promoters of all the rest genes (non-DEGs) https://github.com/ubercomrade/esdeg. The next step was to overlap TFs respecting enriched motifs and DE TFs.
This may point to the most important TFs
A. thaliana motifs are quite good for all cruciferous plants. You needs only promoters of all protein coding genes from the reference genome, that can be extracted conventionally, e.g.
https://plants.ensembl.org/Brassica_oleracea/Info/Index through gff or
https://plants.ensembl.org/biomart/martview.
This issue is may additionally prove the conclusions of the manuscript and motivate concreate further studies concerning specific TFs.
It seems that without this analysis, the title (The molecular mechanism of relatively low temperatures induced broccoli flower bud differentiation revealed by transcriptomic profiling) slightly overestimates the results of study.
All the rest comments are listed below, they are mainly technical.
Figure 2A
I propose apply here the column chart (stacked columns)
Figure 2B
Now areas of almost all Venn diagrams distort the real counts
4 way area-proportional visualization would promoted the representation, e.g.
https://cran.r-project.org/web/packages/eulerr/vignettes/venn-diagrams.html
Line 207
DEGs (P≤0.01 and |log2Ratio|≥1)
I suspect value P here implies an adjusted p-value, this should be mentioned in the manuscript
209-212
This list may be omitted in the text if Fig. 2A would be corrected as described above, and the numbers would be on the stacks of columns
224
Dynamism -> dynamics
Fig. 4 representation may be misleading since only numbers of DEGs are shown, but also it very important how many genes each GO term includes, see your criteria (line 207), adjusted p-values and fold changes reflect the importance of GO terms, see dotplots here, https://learn.gencore.bio.nyu.edu/rna-seq-analysis/gene-set-enrichment-analysis/
Yu G, Wang LG, He QY. ChIPseeker: an R/Bioconductor package for ChIP peak annotation, comparison and visualization. Bioinformatics. 2015; 31: 2382–3.
See examples here, https://www.frontiersin.org/articles/10.3389/fimmu.2021.676173/full
Various GO terms may represent extremely different numbers of genes in whole genomes (up to orders of magnitudes). Consequently, any representation of GO terms should try to combine the abundance and enrichment.
Fig.5 does not explain rich factor and qvalue, I suspect that qvalue means adjusted p-value. If it is so, then qvalue < 0.05 is significant results, qvalue < 0.1 is boundary conditions, but why the limit is extended up to qvalue < 0.4? Why fixed 20 top KEGG pathways are present in each panel?
Fig. 7.
Correct the classification of TF families according to
https://www.biorxiv.org/content/10.1101/2022.11.22.517060v1.full
Blanc-Mathieu R, Dumas R, Turchi L, Lucas J, Parcy F. Plant-TFClass: a structural classification for plant transcription factors. Trends Plant Sci. 2023; S1360-1385(23), 00227-3.
The term class is better here, see
1. Wingender E. Criteria for an updated classification of human transcription factor DNA-binding domains. J Bioinform Comput Biol. 2013;11: 1340007.
2. Wingender E, Schoeps T, Dönitz J. TFClass: an expandable hierarchical classification of human transcription factors. Nucleic Acids Res. 2013;41(Database issue): D165–70.
3. Wingender E, Schoeps T, Haubrock M, Dönitz J. TFClass: a classification of human transcription factors and their rodent orthologs. Nucleic Acids Res. 2015;43(Database issue): D97–102.
4. Wingender E, Schoeps T, Haubrock M, Krull M, Dönitz J. TFClass: expanding the classification of human transcription factors to their mammalian orthologs. Nucleic Acids Res. 2018; 46(D1): D343-7.
5. Castro-Mondragon JA, Riudavets-Puig R, Rauluseviciute I, Lemma RB, Turch L, Blanc-Mathieu R, et al.. JASPAR 2022: the 9th release of the open-access database of transcription factor binding profiles. Nucleic Acids Res. 2022;50(D1): D165–73.
Are raw and processed RNAseq data available in GEO? If not I propose to publish them.
As concern the beginning of discussion, how often differences in the transition to flower bud differentiation are observed for various temperature conditions for other plants? Are there any previous RNA-seq studies on the A. thaliana, its ecotypes or close relatives concerning the influence of temperature condition on the transition of plants to flower bud differentiation? This would be the strong support of the results of the current study.
Comments on the Quality of English LanguageI think that English is satisfactory, but it is still far from very good
Author Response
We have revised the manuscript after careful consideration of the comments made by the three reviewers and editor. Here are our responses in detail. The revised parts used Microsoft Word's built-in track changes function to highlight any changes.
- The merit of the authors is the analysis of transcription factor genes among differentially expressed genes.
The manuscript tries to find among differentially expressed genes (DEGs) those directly involved in transcription regulation, namely the genes of transcription factors. Notably, the conditions of RNA-seq experiment, in particular, variation in transition to the flower bud differentiation proposes that some difference in transcription regulation is expected.
Overall, the analysis of motifs of transcription factor binding sites in DEGs can substantially support the study, in particular, its main results from Fig. 7, 8. Conventional motif enrichment analysis may be performed, but application of special tool developed for analysis of RNA-seq data would be more beneficial. In particular, I propose to test whether motifs of DE TFs genes (Figure 8) are enriched in promoters of all DEGs, see example of such analysis here
Oshchepkov et al. Transcription Factors as Important Regulators of Changes in Behavior through Domestication of Gray Rats: Quantitative Data from RNA Sequencing. Int. J. Mol. Sci. 2022, 23, 12269. https://doi.org/10.3390/ijms232012269.
In this study the tool ESDEG was specially designed to detect the most reliable significantly enriched motifs of known TFs from public library (Plant Cistrome) comparing between promoters of DEGs and promoters of all the rest genes (non-DEGs) https://github.com/ubercomrade/esdeg. The next step was to overlap TFs respecting enriched motifs and DE TFs.
This may point to the most important TFs
- thaliana motifs are quite good for all cruciferous plants. You needs only promoters of all protein coding genes from the reference genome, that can be extracted conventionally, e.g.
https://plants.ensembl.org/Brassica_oleracea/Info/Index through gff or
https://plants.ensembl.org/biomart/martview.
This issue is may additionally prove the conclusions of the manuscript and motivate concreate further studies concerning specific TFs.
It seems that without this analysis, the title (The molecular mechanism of relatively low temperatures induced broccoli flower bud differentiation revealed by transcriptomic profiling) slightly overestimates the results of study.
Response: In this study, the broccoli (Brassica oleracea L. var. italica ) cultivar Zhongqing 10 was selected as the research material. The genome sequence of broccoli has been sequenced, but it has not been published. Broccoli is a variety of Brassica oleracea, and the genome of Brassica oleracea var. oleracea has been published. Therefore, the genome of Brassica oleracea var. oleracea was used as the reference genome. There is no problem analyzing the coding region of broccoli using Brassica oleracea var. oleracea genome as reference genome. However, the analysis of motifs of transcription factor binding sites in DEGs may get wrong results. In addition, although Arabidopsis and Chinese cabbage are cruciferous plants, the gene promoter sequence changes are very large, so the genome sequence of non-native species cannot be analyzed for promoter sequence.
- Figure 2A I propose apply here the column chart (stacked columns)
Figure 2B Now areas of almost all Venn diagrams distort the real counts 4 way area-proportional visualization would promoted the representation, e.g.
https://cran.r-project.org/web/packages/eulerr/vignettes/venn-diagrams.html
Response: Firstly, The column chart (stacked columns) is a very good form of representation, we have made modifications to Figure 2A according to your suggestion. Secondly, figure 2B is similar to the venn-diagrams on the website you recommend. So, figure 2B is still retained.
- Line 207
DEGs (P≤0.01 and |log2Ratio|≥1)
I suspect value P here implies an adjusted p-value, this should be mentioned in the manuscript.
Response: Your suggestion is very reasonable. We have made additions and modifications in section 2.4.
- 209-212
This list may be omitted in the text if Fig. 2A would be corrected as described above, and the numbers would be on the stacks of columns
Response: This list on line 209-212 has been omitted in revised manuscript.
- 224
Dynamism -> dynamics
Response: We carefully checked and revised the spelling of the words in the whole manuscript.
- Fig. 4 representation may be misleading since only numbers of DEGs are shown, but also it very important how many genes each GO term includes, see your criteria (line 207), adjusted p-values and fold changes reflect the importance of GO terms, see dotplots here, https://learn.gencore.bio.nyu.edu/rna-seq-analysis/gene-set-enrichment-analysis/
Yu G, Wang LG, He QY. ChIPseeker: an R/Bioconductor package for ChIP peak annotation, comparison and visualization. Bioinformatics. 2015; 31: 2382–3.
See examples here, https://www.frontiersin.org/articles/10.3389/fimmu.2021.676173/full
Various GO terms may represent extremely different numbers of genes in whole genomes (up to orders of magnitudes). Consequently, any representation of GO terms should try to combine the abundance and enrichment.
Response: Gene Set Enrichment Analysis (GSEA) is a computational method that determines whether a pre-defined set of genes (ex: those beloging to a specific GO term or KEGG pathway) shows statistically significant, concordant differences between two biological states. Figure 4A showed the GO enrichment analysis of DEGs in Analysis Group 1 (V0 compared withV8, V14, V15 and V22), while Figure 4B showed the GO enrichment analysis of DEG in Analysis Group 2 (CK compared to V14 and V15). There were five and three biological states in Figures 4A and 4B, respectively. Therefore, Figure 4 cannot be modified according to your suggestions.
- Fig.5 does not explain rich factor and qvalue, I suspect that qvalue means adjusted p-value. If it is so, then qvalue < 0.05 is significant results, qvalue < 0.1 is boundary conditions, but why the limit is extended up to qvalue < 0.4? Why fixed 20 top KEGG pathways are present in each panel?
Response: Each circle in the figure 5 represents a KEGG pathway, the ordinate represents the pathway name, and the abscissa represents Enrichment Factor (Enrichment Factor), which represents the ratio of the proportion of DEGs annotated to a pathway to the proportion of all genes annotated to that pathway. The greater the enrichment factor indicates the more significant the enrichment level of DEGs in this pathway. The color of the circle represents qvalue, and qvalue means the adjusted p value. The smaller qvalue suggest the more reliable the enrichment significance of DEGs genes in this pathway. The size of the circle indicates the number of genes enriched in the pathway, and the larger the circle represent the more genes. As you stated, qvalue < 0.05 is significant results, qvalue < 0.1 is boundary conditions. However, in KEGG analysis, it is not particularly emphasized that qvalue < 0.05, e.g. https://www.sciencedirect.com/science/article/abs/pii/S0308814619317546?via%3Dihub
LiuYH, Lv JH, Liu ZB, et al.. Integrative analysis of metabolome and transcriptome reveals the mechanism of color formation in pepper fruit (Capsicum annuum L.). Food Chem.2020, 306, 125629. https://doi.org/10.1016/j.foodchem.2019.125629.
- Fig. 7.
Correct the classification of TF families according to
https://www.biorxiv.org/content/10.1101/2022.11.22.517060v1.full
Blanc-Mathieu R, Dumas R, Turchi L, Lucas J, Parcy F. Plant-TFClass: a structural classification for plant transcription factors. Trends Plant Sci. 2023; S1360-1385(23), 00227-3.
The term class is better here, see
- Wingender E. Criteria for an updated classification of human transcription factor DNA-binding domains. J Bioinform Comput Biol. 2013;11: 1340007.
- Wingender E, Schoeps T, Dönitz J. TFClass: an expandable hierarchical classification of human transcription factors. Nucleic Acids Res. 2013;41(Database issue): D165–
- Wingender E, Schoeps T, Haubrock M, Dönitz J. TFClass: a classification of human transcription factors and their rodent orthologs. Nucleic Acids Res. 2015;43(Database issue): D97–
- Wingender E, Schoeps T, Haubrock M, Krull M, Dönitz J. TFClass: expanding the classification of human transcription factors to their mammalian orthologs. Nucleic Acids Res. 2018; 46(D1): D343-7.
- Castro-Mondragon JA, Riudavets-Puig R, Rauluseviciute I, Lemma RB, Turch L, Blanc-Mathieu R, et al.. JASPAR 2022: the 9th release of the open-access database of transcription factor binding profiles. Nucleic Acids Res. 2022;50(D1): D165–
Response: We correct the classification of TF families in figure 7 and the whole manuscript according to Blanc-Mathieu R.
- Are raw and processed RNA-seq data available in GEO? If not I propose to publish them.
Response: We have not uploaded the raw and processed RNA-seq data to GEO, and we currently have no plans to upload the data to GEO. But, the data will be uploaded to GEO at a suitable time in the future.
- As concern the beginning of discussion, how often differences in the transition to flower bud differentiation are observed for various temperature conditions for other plants? Are there any previous RNA-seq studies on the A. thaliana, its ecotypes or close relatives concerning the influence of temperature condition on the transition of plants to flower bud differentiation? This would be the strong support of the results of the current study.
Response: Your suggestion are very reasonable. Firstly, at the beginning of discussion, we have added that low temperature is necessary for FBD of Arabidopsis late-flowering ecotypes, Chinese cabbage and radish. Secondly, in the foundation of referring to a great deal of literature, there were no reports of RNA-seq studies on A. thaliana, its ecotypes or close relatives concerning the influence of temperature condition on the transition of plants to flower bud differentiation.
- I think that English is satisfactory, but it is still far from very good.
Response: We carefully checked and revised the spelling of the words and grammar in the whole manuscript.
Reviewer 3 Report
Comments and Suggestions for Authors
Chai et al., monitored the morphology and transcriptomic changes of broccoli shoot apices development under low and control temperature conditions to study factors involve in broccoli flower bud differentiation. A comprehensive transcriptomics analyses were conducted which could potentially provide resource for studying cruciferous plants flower bud differentiation. However the paper cannot be accept at current stage. The major concern is lacking connection between the transcriptomic study within low temperature time points and the study in CK(higher temperature) and low temperature of this paper.
comments
1. It's not clear in the paper for the purpose of conducting two sets of transcriptomics analyses between different time points at low temperature and between the CK and low temperature. I am worry about the DEGs, especially the hormone and TFs between CK and low temperature were mainly response to cold stress rather than FBD. Please discuss the concept of experimental design and purpose of the two sets of RNAseq comparisons.
2. following the point 1, the link between comparisons within different time points under low temperature and the comparison between CK and low temperature is lacking. Did the author find overlapping genes in Figure 3A and Figure3B? I also hope the author can discuss the different enriched GO or KEGG pathway terms between the two comparison sets in section 3.4.
3. NO RNAseq and related bioinformatics analyses sections in Material & Method.
4. Please provide detailed description for the data/gene set used for studying. For example, in section 3.5, 3.6, and 3.7, where the DEGs of hormone, TF, and FBD come from? from which comparison pairs?
Comments on the Quality of English Language
English quality is basically acceptable except for a few grammar issue. Describe data more clearly is desired for this manuscript.
Author Response
We have revised the manuscript after careful consideration of the comments made by the three reviewers and editor. Here are our responses in detail. The revised parts used Microsoft Word's built-in track changes function to highlight any changes.
- It's not clear in the paper for the purpose of conducting two sets of transcriptomics analyses between different time points at low temperature and between the CK and low temperature. I am worry about the DEGs, especially the hormone and TFs between CK and low temperature were mainly response to cold stress rather than FBD. Please discuss the concept of experimental design and purpose of the two sets of RNA-seq comparisons.
Response: The broccoli shoots tips treated at relatively low temperatures for 0 d, 8 d, 14 d, 15 d, and 22 d and under the control condition for 15 d were sampled and designated V0, V8, V14, V15, V22 and CK, respectively. In this study, FBD began at 15 d in the treatment group (17°C/9°C), while in the control group (25°C/17°C) the broccoli shoot apices remained in the vegetative growth stage. So, the DEGs between CK and low temperature were mainly response to FBD.
- following the point 1, the link between comparisons within different time points under low temperature and the comparison between CK and low temperature is lacking. Did the author find overlapping genes in Figure 3A and Figure3B? I also hope the author can discuss the different enriched GO or KEGG pathway terms between the two comparison sets in section 3.4.
Response: We analyzed overlapping genes in Figure 3A and Figure3B and found no correlation with flower bud differentiation, so they were not presented in the manuscript. In addition, RNA-seq technology was used to analyze the DEGs after different durations of treatment at relatively low temperatures to screen the key genes involved in the regulation of broccoli FBD. In the manuscript, broccoli FBD was more important, and low temperature was only a condition for broccoli FBD. Therefore, the different enriched GO or KEGG pathway terms between the two comparison sets in section 3.4 were not analyzed.
- NO RNA-seq and related bioinformatics analyses sections in Material & Method.
Response: 2.4 RNA extraction and gene expression profiling on line138 has been changed to 2.4 RNA-Seq and Differential Expression Analysis. RNA-seq and related bioinformatics anslyses has been added in section 2.4.
- Please provide detailed description for the data/gene set used for studying. For example, in section 3.5, 3.6, and 3.7, where the DEGs of hormone, TF, and FBD come from? from which comparison pairs?
Response: Your proposal is reasonable. We have added detailed description in section 3.5 (line 314), 3.6 (line 338-339), and 3.7 (line 380-381).
- English quality is basically acceptable except for a few grammar issue. Describe data more clearly is desired for this manuscript.
Response: We carefully checked and revised the spelling of the words and grammar in the whole manuscript.
Round 2
Reviewer 2 Report
Comments and Suggestions for Authors
I am not quite satisfied with the author’s reply, while the first impression of the manuscript was substantially better. Only minor and quite technical corrections are seen in the revised version of the manuscript, while the major issues are almost completely ignored. Meanwhile, I did not proposed additional experiments that would require extra time and resources. I proposed only bioinformatics analysis, which can be done very fast and promote overall value of the paper.
1
I am still insist on the analysis of transcription factor binding sites in DEGs. There are two options here. First, this may be performed for Brassica oleracea var. oleracea since authors used it as the reference genome, and though DNA in non-coding regions are evolved faster than in coding ones, this analysis propose only overall density of motif’s concurrencies, which hardly can change very fast in evolution. The second option is here http://www.bogdb.com/genome/broccoli, Wang Y, Ji J, Fang Z, Yang L, Zhuang M, Zhang Y, Lv H. BoGDB: An integrative genomic database for Brassica oleracea L. Front Plant Sci. 2022 Aug 24;13:852291. doi: 10.3389/fpls.2022.852291. There authors can get access to the upstream regions of genes, whole genome, actually, is not required. If somehow this species is not correct, try to use internet search to find something else since I used a couple of minute to find the link above.
2
Areas in Fig. 2B-E are still not proportional
3
Fig. 4 representation may be misleading since only numbers of DEGs are shown, but for various GO terms distinct numbers of genes are observed in whole genome, hence the current representation combines specific and non-specific enrichment. Use standard metrics like fold, p-value, etc.
4
Fig.5. It is not important what anyone done before, even in journals Nature and Cell false results are possible. So please represent data uniformly in different panels, i.e. use the same threshold for qvalue, since now the same color means different qvalues in various panels. This is wrong way to present your data
5
I am not understand why authors refuse to publish their RNA-seq data in GEO or anywhere else. This only means that their data or their processing are not good.
Comments on the Quality of English Language
from moderate to good
Author Response
The molecular mechanism of relatively low temperatures induced broccoli flower bud differentiation revealed by transcriptomic profiling
We have revised the manuscript after careful consideration of the comments made by the two reviewers and editor. Here are our responses in detail. The revised parts used Microsoft Word's built-in track changes function to highlight any changes.
Reviewer #2
- I am still insist on the analysis of transcription factor binding sites in DEGs. There are two options here. First, this may be performed for Brassica oleracea Oleracea since authors used it as the reference genome, and though DNA in non-coding regions are evolved faster than in coding ones, this analysis propose only overall density of motif’s concurrencies, which hardly can change very fast in evolution. The second option is here http://www.bogdb.com/genome/broccoli, Wang Y, Ji J, Fang Z, Yang L, Zhuang M, Zhang Y, Lv H. BoGDB: An integrative genomic database for Brassica oleracea L. Front Plant Sci. 2022 Aug 24;13:852291. doi: 10.3389/fpls.2022.852291. There authors can get access to the upstream regions of genes, whole genome, actually, is not required. If somehow this species is not correct, try to use internet search to find something else since I used a couple of minute to find the link above.
Response: We tried our best to learn the use of ESDEG tools, and analyzed DE TF(Figure 7 and Figure 8) binding sites in DEGs from CK compared with V14 and V15.
Since the Arabidopsis thaliana motifs are quite good for all cruciferous plants, motifs of known TFs were compiled from the JASPAR database with Arabidopsis thaliana. Therefore, the section 2.5 and 3.8 and Figure 9 were added to the manuscript.
- Areas in Fig. 2B-E are still not proportional.
Response: Fig.2 has been changed according to reviewer’s suggestion. When redrawing the figure, we found some errors in the numbers and made corrections.
- Fig. 4 representation may be misleading since only numbers of DEGs are shown, but for various GO terms distinct numbers of genes are observed in whole genome, hence the current representation combines specific and non-specific enrichment. Use standard metrics like fold, p-value, etc.
Response: Fig.4 has been changed according to reviewer’s suggestion.
- Fig.5. It is not important what anyone done before, even in journals Nature and Cell false results are possible. So please represent data uniformly in different panels, i.e. use the same threshold for qvalue, since now the same color means different qvalues in various panels. This is wrong way to present your data
Response: There was a little wrong of Figure 5. We have changed it according to reviewer’s suggestion.
- I am not understand why authors refuse to publish their RNA-seq data in GEO or anywhere else. This only means that their data or their processing are not good.
Response: We have published the RNA-seq data in NCBI (PRJNA1022574). Because our manuscript has not been published online, our data is currently not publicly available.
Reviewer 3 Report
Comments and Suggestions for Authors
The authors basically addressed my concerns. Could the authors merge the answers of point 1 and point 2 from my 1st round of comments into the proper sections of the manuscript (result, discussion, etc.)? I still hope the manuscript could discuss the purpose/design of experiments in this study of having two sets of RNAseq.
Comments on the Quality of English LanguageIt's acceptable in current condition
Author Response
The molecular mechanism of relatively low temperatures induced broccoli flower bud differentiation revealed by transcriptomic profiling
We have revised the manuscript after careful consideration of the comments made by the two reviewers and editor. Here are our responses in detail. The revised parts used Microsoft Word's built-in track changes function to highlight any changes.
1. The authors basically addressed my concerns. Could the authors merge the answers of point 1 and point 2 from my 1st round of comments into the proper sections of the manuscript (result, discussion, etc.)? I still hope the manuscript could discuss the purpose/design of experiments in this study of having two sets of RNA-seq.
Response: Thank you for the affirmation. The answer of point 1 from your 1st round of comments can be found in section 3.1. In addition, the answer of point 2 from your 1st round of comments was emerged into section 3.3. And the purpose of experiments in this study of having two sets of RNA-seq has been described in section 3.3.